# Health and Economic Impact Assessment of Transport and Industry PM$_{2.5}$ Control Policy in Guangdong Province

Songyan Ren [1], Peng Wang [1,*], Hancheng Dai [2], Daiqing Zhao [1] and Toshihiko Masui [3]

1    Guangzhou Institute of Energy Conversion, Chinese Academy of Sciences, Guangzhou 510640, China; rensy@ms.giec.ac.cn (S.R.); zhaodq@ms.giec.ac.cn (D.Z.)
2    College of Environmental Sciences and Engineering, Peking University, Beijing 100871, China; dai.hancheng@pku.edu.cn
3    National Institute for Environmental Studies, Tsukuba 300-4352, Japan; masui@nies.go.jp
*    Correspondence: wangpeng@ms.giec.ac.cn

**Abstract:** PM$_{2.5}$ pollution-related diseases lead to additional medical expenses and the loss of working hours, thus affecting the macro-economy. To evaluate the health-related economic impacts of PM$_{2.5}$, the Integrated Assessment Model of Climate, Economic, and Environment (ICEEH), combined with the Computable General Equilibrium (CGE) model, the Greenhouse Gas and Air Pollution Interactions and Synergies (GAINS) model, and a health impact assessment module was constructed. The impact of different air pollution control strategies was analyzed in Guangdong Province by establishing a Without Control (WOC) scenario, an Air Control (AIC) scenario, and a Blue Sky (BLK) scenario. The results show that in the WOC scenario for 2035, the death rate for Guangdong Province is 71,690 persons/year and the loss of working hours is 0.67 h/person/year. In the AIC and BLK scenarios compared with WOC for 2035, the loss of working hours is reduced by 29.8% and 34.3%, and premature deaths are reduced by 33.0% and 37.5%, respectively; GDP would increase by 0.05% and 0.11%, respectively, through strict pollution control policies. Furthermore, improved labor force quality induced by better air conditions would promote the added value in labor-intensive industries, such as agriculture (0.233%), other manufacturing (0.172%), textiles (0.181%), food (0.176%), railways transport (0.137%), and services (0.129%). The added value in the waste (−0.073%), nature gas (−0.076%), and crude oil sectors (−0.072%) would decrease because of the increased investment installment in PM$_{2.5}$ treatment equipment.

**Keywords:** health impact; PM$_{2.5}$ pollution; transport and industry; air control strategy

## 1. Introduction

As a developing country, China is faced with the dual challenges of promoting both economic development and environmental protection. In the process of comprehensively promoting modernization, China regards environmental protection as a basic national policy and sustainable economic development as an important strategy. China carries out pollution prevention and ecological environmental protection activities nationwide. In 2016, China submitted the goal of its independent national contribution (INDC) at the Conference of the Parties 21 (COP 21) in Paris. China formally promised that $CO_2$ emissions will peak around 2030, it will achieve carbon neutrality around 2060, and it will strive to reach this peak as soon as possible; $CO_2$ intensity will decrease by 60% to 65% compared with 2005, and non-fossil energy will account for about 20% of primary energy consumption in 2030. In particular, China announced further commitments for 2030 at the 2020 Climate Ambition Summit, promising to lower its carbon dioxide emissions per unit of GDP by over 65 percent from the 2005 level, increase the share of non-fossil fuels in primary energy consumption to around 25 percent, increase the forest stock volume by six billion cubic meters from the 2005 level, and increase its total installed capacity of wind and solar power to over 1.2 billion kW [1]. Addressing climate change and protection of the environment

progress simultaneously. In 2018, the State Council issued a three-year action plan to win the blue sky defense war, which proposed that after three years of effort, total emissions of major air pollutants should be significantly reduced, greenhouse gas emissions should be reduced in coordination, concentrations of fine particulate matter ($PM_{2.5}$) should be further significantly reduced, the number of days of heavy pollution should be significantly reduced, environmental air quality should be significantly improved, and the quality of life should be significantly enhanced.

As the largest province in terms of GDP and a low-carbon pilot province in China, Guangdong should be at the forefront of the country in air pollution control. In 2017, the GDP of Guangdong Province reached CNY (Chinese Yuan) 8.97 trillion and energy consumption was 317 million tons of standard coal, both ranking first in China. However, the average annual concentration of $PM_{2.5}$ was 33 μg/m$^3$ for Guangdong in 2017, far higher than the standard of class I of 10 μg/m$^3$, and class II of 25 μg/m$^3$, set by the World Health Organization (WHO). Therefore, there is significant room for improvement. In May 2018, the Guangdong Provincial Government proposed an implementation plan for winning the blue sky defense war in Guangdong Province (2018–2020). The plan points out that, by 2020, the proportion of days with an air quality compliance index (AQI) in the province will reach 92.5%, annual average concentration of fine particles ($PM_{2.5}$) will be controlled below 33 μg/m$^3$, and heavy pollution will be basically eliminated. The total emissions of sulfur dioxide ($SO_2$) and nitrogen dioxide ($NO_2$) in the province will have decreased by 5.4% and 3%, respectively, compared with 2015, yet no specific emission reduction indicators were issued for $PM_{2.5}$. According to the report on the state of the Guangdong Provincial Ecology and Environment in 2018, the primary pollutant was $O_3$ (59.6%), followed by $PM_{2.5}$ (21.5%) and $NO_2$ (10.6%), which shows that $PM_{2.5}$ and related pollutants are very serious in Guangdong [2].

Specifically, $PM_{2.5}$ pollution may lead to acute and chronic diseases, generate additional health expenditure, and affect the attendance rate of the labor force, thus shortening working hours. These health problems will cause a heavy economic burden by further increasing health expenditure, increasing workday loss, and reducing labor supply. China's current fertility rate is 1.5–1.6, close to the level of the developed countries. At the same time, the aging of the population is gradually increasing, so the demand for labor quality will gradually increase. However, air pollution may lead to a decrease in the supply of labor and negatively impact the economy. A study in the United States found that ozone ($O_3$) levels well below federal air quality standards had a significant impact on the productivity of agricultural workers [3]. An experiment in Mexico City showed that a 19.7% reduction in sulfur dioxide resulted in an increase of 1.3 h (or 3.5%) per week [4]. In China, the economic impact of air pollution accounts for 0.72–6.94% of regional GDP, according to several city and provincial studies [5–7]. In 2013, the global welfare loss caused by outdoor air pollution exceeded USD (United States dollar) 5 trillion [8]. The estimation of these losses is mostly based on local survey data, and econometric methods are often used, such as willing to pay (WTP), value of a statistical life (VSL), human capital (HCA), and cost of illness (COI) [9,10]. The existing environmental health benefit assessments have mostly studied the emission reduction in pollutants caused by pollutant treatment according to monitoring data or air quality models, then used epidemiology to study the impact of air quality improvement on human health, and finally carried out a health benefit assessment according to the willingness to pay and disease cost methods. Due to the failure to consider the pollutant reduction and environmental health benefits resulting from the adjustment of the front-end energy structure, industrial structure, and transportation structure, these previous studies are subject to limitations, and it is impossible to evaluate the environmental health benefits resulting from energy transformation as the carbon constraint, such as the INDC target.

Other scholars have used the Computable General Equilibrium (CGE) model as a systematic tool to measure the economic impact of air pollution [11–14]. Based on the CGE linkage model of the OECD, the economic loss caused by global outdoor air pollution is expected to reach 1% of global GDP [12]. In addition, the Air Quality model was used to

assess health externalities to support air quality plan and action [13]. China's GDP losses caused by air pollution will reach 2.6% of GDP by 2060 [14]. Another study about the impact of air pollution on China's economy found losses of USD 50 billion in 2000 and USD 115 billion in 2005 [10]. China will experience a GDP loss of 2% and a cost of USD 25.2 billion in medical expenditures in 2030 related to $PM_{2.5}$ pollution [15]. Wu et al. [16] analyzed the improvement in air health effects resulting from the reduction in local air pollution, deriving an amount of 1.01% of the local GDP due to the control of fossil energy in Shanghai. At present, most of the literature is based on the welfare loss caused by air pollutants, but few studies have systematically modeled and analyzed the impact of pollutant emission reductions on air quality under different air quality control policy scenarios, and then analyzed the impact of air quality after pollutant reductions on health (including medical treatment, loss of work, and early death), so as to further evaluate the impact of health on the macro-economy (resulting in loss of work and medical expenditure).

Previous studies generally focused on northern China, where air pollution is more serious, and less on coastal areas, which have slightly better air quality. Some studies evaluated welfare loss, which does not consider the cost and benefit of pollutant prevention linked with the macro-economy. However, Guangdong Province is located in southern China, adjacent to the sea, and the pollutant concentration is less serious than that in the north.

However, in recent years, the air quality requirements have been benchmarked with international big cities to promote the development of the environmental protection industry and improve the quality of people's health through strict air quality standards. The cost and effectiveness of pollutant control policies over the years need to be carefully evaluated.

In view of the above research deficiencies, this study constructed a comprehensive evaluation model of Climate, Energy & Economic & Health (ICEEH) in Guangdong by linking an energy economy model, an air quality model, and a health risk model. The aim was to address the data requirements of the air quality model for the energy structure and energy consumption, and the health risk assessment model for pollutant emission and pollutant concentration. It also address the problem that the energy economic model does not consider environmental externalities, resulting in overestimation of transformation costs. By designing different air control policy scenarios, the comprehensive impact of the air control strategy caused by air control technology and energy transformation, constrained by INDC toward 2035, on the economy, the environment, and people's health in Guangdong was systematically evaluated.

Logically, investing in emissions reduction technology for $PM_{2.5}$ will incur high costs, However, in making such decisions, the government must consider the costs that society will pay under declining levels of different air quality standards and, in particular, the impact on the health of the labor force and economic output. Although improving air quality will increase the cost of the emissions reduction equipment, it is possible that pollution control has a significant effect of reducing health expenditure, improving health levels, and increasing disposable income. The purpose of this study was to prove this hypothesis through the case study of Guangdong and further promote the applicability of the overall evaluation model. Therefore, it is necessary to synergistically analyze the cost of emissions reduction technology and the improvement in the macro-economy. By setting different air pollution control strategy scenarios, the $PM_{2.5}$ emissions from different sectors in Guangdong Province were identified, facilitating quantitative analysis of the impact on the energy, economy, environment, and human health, comprehensively assessing the impact of the emissions reduction and health on the macro-economy, and providing policy support for policy-making regarding highly polluting industries.

## 2. Methodology

To evaluate the impact of $PM_{2.5}$ pollutant emission reduction on health and the economy, the Integrated Assessment Model of Climate, Energy & Economic & Health (ICEEH) was built, which is composed of the CGE model [17], the Greenhouse Gas-Air Pollution

Interactions and Synergies (GAINS) model [18], and the health impact module with the exposure effect equation [19], as shown in Figure 1. The CGE model was used to predict the future energy demand under the different air quality improvement scenarios and link the labor force with GDP. The GAINS model was used to calculate annual average $PM_{2.5}$ emissions, concentrations, and pollution control costs for Guangdong Province. Using the exposure population, $PM_{2.5}$ concentration, and exposure effect equation, the number of pathogenic cases and premature deaths caused by $PM_{2.5}$ pollution were calculated. The additional health expenditure and decrease in the labor force time supply were estimated. These factors were then input into the CGE model to evaluate the impact on the regional economy and welfare.

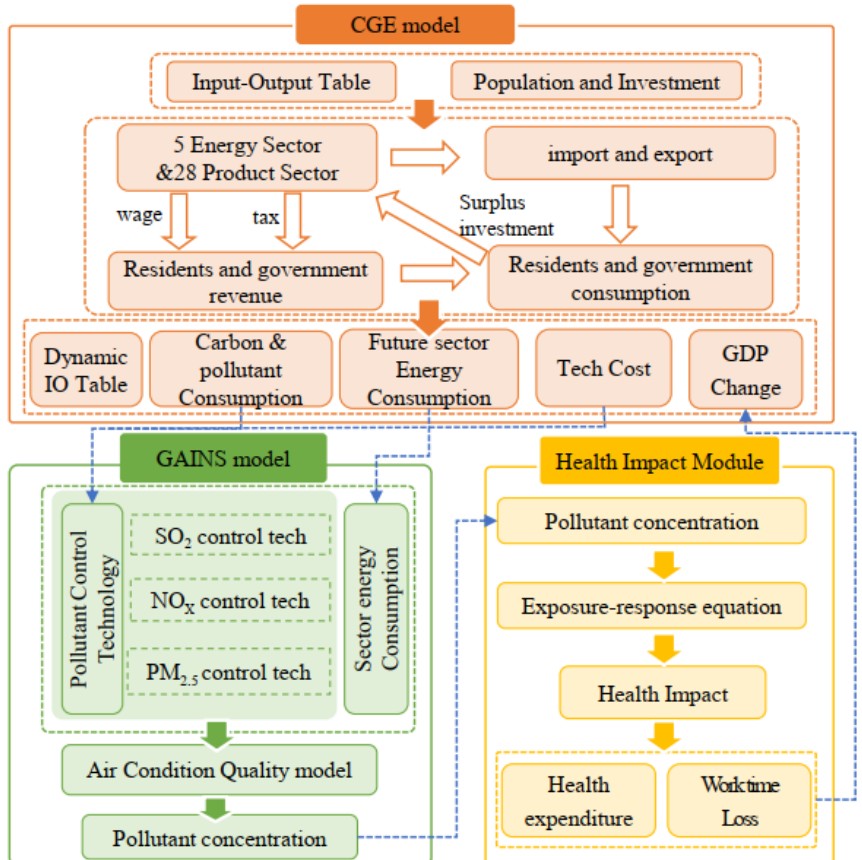

**Figure 1.** ICEEH model structure.

## 2.1. Integrated Model Structure

### 2.1.1. CGE Model

The CGE model is a practical application based on the general equilibrium theory of economics, which considers the economic system as a whole. Due to the spillover effect between markets, the general equilibrium requires all markets to clear and reach equilibrium at the same time. Therefore, the modeling process of CGE model is actually to change Wallace's general equilibrium theory from an abstract form to a practical model. Under the ideal general equilibrium, consumers realize the maximization of utility and enterprises realize the maximization of profit, so as to solve the equilibrium price and quantity that can clear all markets at the same time. The CGE model is disturbed under a scenario setting, and the balance will be regained in the future development scenario. When the economic system returns to the balanced state, the impact of changes in various economic variables can be recalculated. This study quantified the increase in medical expenditure and the decrease in labor force induced by the delay time and premature deaths caused by air pollution, and evaluated the economic loss through the CGE model. The

CGE model is a recursive dynamic model based on the input–output table of Guangdong Province in 2007 [20]. There are 33 production sectors in the model, including seven energy sectors (see Table 1). All sectors use constant elasticity of substitution (CES) for their activities. The input parameters include intermediate products, energy commodities, initial labor force, and capital. Based on the energy balance sheet, energy commodities are divided into material use and fuel use.

**Table 1.** Classification of production sectors.

| No. | Sector Name | No. | Sector Name |
|---|---|---|---|
| 1 | Agriculture | 18 | Steel |
| 2 | Coal mining | 19 | Non-ferrous metal smelting |
| 3 | Petroleum mining | 20 | Metalware |
| 4 | Natural gas mining | 21 | Mechanical manufacturing |
| 5 | Other mining | 22 | Electronic equipment manufacturing |
| 6 | Food manufacturing | 23 | Power generation |
| 7 | Textile | 24 | Gas production and supply |
| 8 | Wood processing | 25 | Water production and supply |
| 9 | Papermaking | 26 | Construction |
| 10 | Other manufacturing | 27 | Road transport |
| 11 | Oil refining | 28 | Railway transportation |
| 12 | Coking | 29 | Urban public transport |
| 13 | Chemical industry | 30 | Water transport |
| 14 | Cement | 31 | Air transport |
| 15 | Other non-metallic manufacturing industry | 32 | Other transportation |
| 16 | Glass manufacturing | 33 | Service industry |
| 17 | Ceramic manufacturing | | |

The model is solved at one-year time steps towards 2030 in a recursive-dynamic manner, in which the selected variables, including capital stock, labor force, land, natural resource, energy efficiency improvement (EEI), total factor productivity (TFP), land productivity, and extraction cost of fossil fuels, are updated based on the modeling of intertemporal behavior and results of previous periods (Equations (1)–(4)).

Capital accumulation process:

$$I_{r,tot,t+1} = \sum_j CAPSTK_{r,j,t+1} \times \left[ (1 + g_{r,t+1})^T - (1 - d_r)^T \right] \tag{1}$$

$$CAPSTK_{r,j,t+1} = (1 - d_r)^T \times CAPSTK_{r,j,t} + T \times I_{r,i,t} \tag{2}$$

where total investment ($I_{r,tot,t}$) is given exogenously, investment in sector $j$ ($I_{r,j,t}$) is determined by the model depending on the rate of return to capital, capital stock accumulation ($CAPSTK_{r,j,t}$), $g_{r,t+1}$ is the expected GDP growth rate in the year $t + 1$, $d_r$ is the depreciation rate (5% for all regions), and $T$ is time step (1 year).

Supply of total labor force:

$$F^t_{r,pf} = F^{t-1}_{r,pf} \times \left( 1 + gr^t_{r,pf} \right) \tag{3}$$

where $F^t_{r,pf}$ is the primary factor (*pf*) of labor force and $gr^t_{r,pf}$ is the corresponding exogenous growth rate.

Efficiency parameters:

The CGE model distinguishes technological efficiency improvement of new investments from that of existing capital stock.

For new investments, sectoral efficiencies of energy, land productivity and total factor productivity are given as exogenous scenarios, whereas for existing capital stock, efficiency

of *par* (*par* $\in$ efficiency of energy and capital) in time $t$ ($\text{EFF}_{r,par,j}^{ext,t}$) is the average of capital stock ($\text{EFF}_{r,par,j}^{ext,t-1}$) and new investments ($\text{EFF}_{r,par,j}^{new,t-1}$) in the previous period, here:

$$\text{EFF}_{r,par,j}^{ext,t} = \frac{\left(\text{EFF}_{r,par,j}^{ext,t-1} \times CAPSTK_{r,j,t-1} + \text{EFF}_{r,par,j}^{new,t-1} \times I_{r,j,t-1}\right) \times (1 - d_r)^T}{CAPSTK_{r,j,t}} \tag{4}$$

### 2.1.2. GAINS Model

The cost and benefit analysis through the GAINS model uses energy consumption data in different sectors from the CGE model as its input. The GAINS model is used to estimate air pollutant emissions and design emission reduction strategies for different industry sectors. It provides a unified framework to estimate emissions, emission reduction potential, and cost of emissions reduction, including greenhouse gases.

The GAINS model covers the power, heat production and supply, iron and steel, chemical, non-ferrous metal, and transportation industries, along with other light industries and sectors. The model is applied to these sectors to predict the potential for air pollutant and greenhouse gas emission reduction from implementation of air pollution control measures in various industries.

The main input parameters are the energy activity level and pollutant control strategy. The energy activity level includes four modules: power, industry, residential, and transportation. The pollutant control strategy classifies the technologies of transportation, residential, and industrial boilers in detail.

The GAINS model considers air pollutants such as $SO_2$, $NO_x$, $PM_{2.5}$, $PM_{10}$, $NH_3$, and VOC, and greenhouse gases such as $CO_2$, $CH_4$, $N_2O$, CFCs, HFCS, and $SF_6$. Therefore, the GAINS model can be used to calculate the coordination degree of emissions reductions of greenhouse gases and pollution factors under different scenarios and different levels of activities modified according to air pollution control measures. The GAINS model can be used to evaluate the impact of particulate pollution, acidification, eutrophication, and tropospheric ozone on health and the environment.

The GAINS model also considers the emission characteristics of specific regions and industry sources, analyzes the emissions of all major air pollutants and greenhouse gases caused by emissions reduction measures, and simulates the accumulation and dispersion process of emissions in the atmosphere. It also enables the impact of the health level to be estimated.

According to the research purpose, the scenario hypothesis of different emission reduction intensities in the GAINS model can be changed by inputting Guangdong energy data into the GAINS model. Through the model calculation, the results of the change in pollutant emissions under different policies, annual average $PM_{2.5}$ concentration of each province, and the emissions reduction cost under each scenario can be evaluated.

The calculation formula of pollutant discharge is as follows:

$$E_y = \sum_i \sum_j A_i e_{y,\,i} T_j \left(1 - \theta_{y,j}\right) \tag{5}$$

where

$E_y$: Discharge of pollutant $y$

$i$: Fuel type

$j$: Control Technology

$A_i$: Fuel consumption (10,000 tons of standard coal)

$e_{y,\,i}$: Emission factor of fuel $i$ (ton air pollutants per ton fuels)

$T_j$: Technology penetration rate (%);

$\theta_{y,j}$: Removal rate (%)

In the GAINS model, the emission reduction cost per unit pollutant is defined as the total annual cost of emission reduction per unit pollutant, including initial investment costs,

fixed operating costs, and variable operating costs. The control cost per unit pollutant is calculated as follows:

$$ca_{k,j} = \frac{I_{k,j}^{an} + OM_{k,j}^{fix}}{2a} + OM_{k,j}^{var} \tag{6}$$

where

$k$: type of activity

$A_k$: Activity level

$ca_{k,j}$: Unit abatement cost

$I_{k,j}^{an}$: Annual investment cost

$OM_{k,j}^{fix}$: Fixed operating costs

$OM_{k,j}^{var}$: Variable operating costs

The total pollutant control cost $C$ is calculated as follows:

$$C = \sum_k \sum_j A_k ca_{k,j} T_j \tag{7}$$

### 2.1.3. Health Analysis Module

The health analysis module includes the exposure effect coefficient and health risk assessment. The exposure response coefficient is derived from a cohort study of long-term exposure to human health. The Harvard Six Cities Study and the American Cancer Society Cohort Study are widely recognized cohort studies on the relationship between air pollution exposure and human health [21]. The latest research shows that there is a nonlinear relationship between $PM_{2.5}$ concentration and lethality rate. In addition, Chen et al. determined the results of the long-term health effects of $PM_{10}$ and TSP pollution in China, and the results caused by three different exposure effect relationships [22]. To evaluate the incidence of $PM_{2.5}$ pollution-related diseases, the exposure effect coefficient is used [15,23–25]. Due to the lack of such an exposure response parameter in Guangdong, the data of these studies are used for reference.

Health effects refer to the series of health problems caused by exposure to high concentrations of $PM_{2.5}$, including disease and death. The relative risk of $PM_{2.5}$ pollution and health effects demonstrate the relationship between exposure and health effects. Early studies show a linear relationship between the exposure of $PM_{2.5}$ pollution and morbidity and mortality, whereas recent studies show that the relationship is nonlinear [23,26]. When $PM_{2.5}$ concentration is more than 10 μg/m$^3$, the number of premature deaths and the number of all age-old patients caused by $PM_{2.5}$ pollution can be calculated, and premature deaths in those aged 15 to 65 years will reduce the labor force supply. According to the age-specific mortality rate, the number of premature deaths among workers and the number of related diseases caused by $PM_{2.5}$ pollution was obtained, and then the loss of working time was estimated.

The formula for the influence of exposure concentration on healthy terminals is as follows, the relative risk factor $\beta_i$ is shown in Table A1:

$$Y_i = y_i P[\beta_i (C - C_0)] \tag{8}$$

where

$Y_i$: Healthy terminal

$y_i$: Basic morbidity or mortality of healthy terminal $i$

$P$: Size of exposed population

$\beta_i$: The relative risk factor of $i$ is the C-R parameter

$C$: The $PM_{2.5}$ concentration of the scenario

$C_0$: Concentration threshold for health effects

## 2.2. Datebase

### 2.2.1. Current Situation of Energy Consumption and GDP in Guangdong Province

The consumption of terminal energy in Guangdong Province has seen rapid growth. As shown in Figure 2, from 2000 to 2017 [27], the annual growth rate of energy consumption in Guangdong Province reached 7.6%. The growth rate of energy consumption remained above 10% until 2007, dropping to 3–5% after the Global Financial Crisis (GFC) of 2008. Economic growth leads to greater energy consumption. With increased energy consumption, emissions of $PM_{2.5}$ increase, impacting the health level.

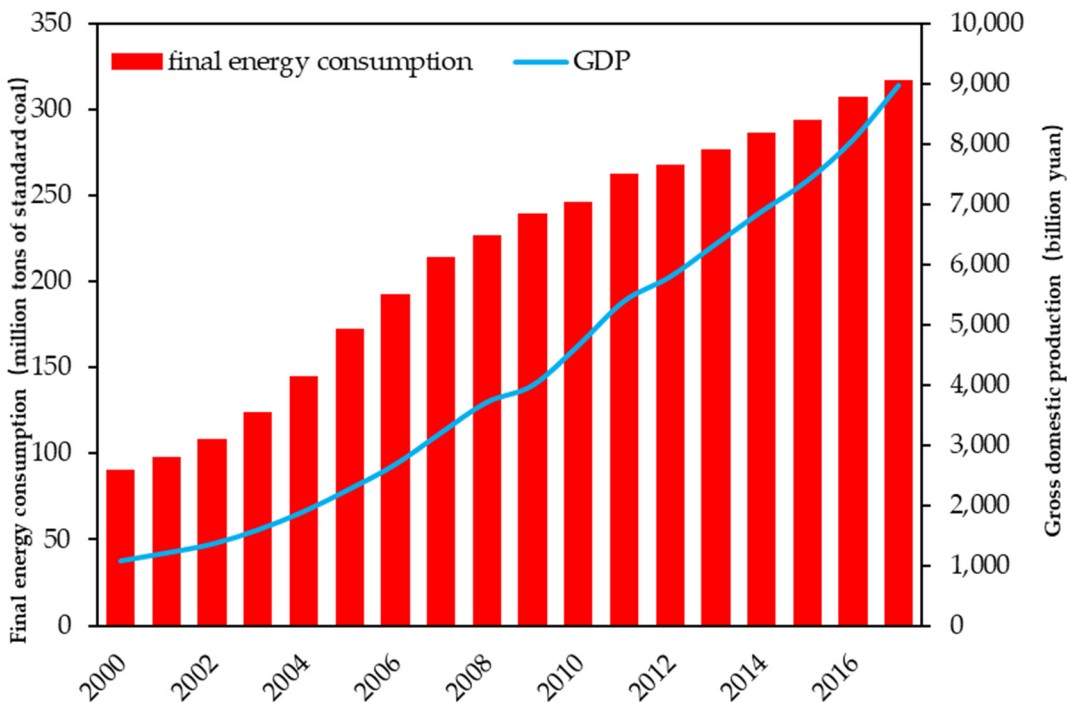

**Figure 2.** Final energy consumption and GDP in Guangdong.

### 2.2.2. $PM_{2.5}$ Emissions in Guangdong Province

According to the Atmospheric Composition Analysis Group of the United States and the Environmental Statistical Bulletin of Guangdong Province, the Atmospheric Composition Analysis Group uses NASA satellite data and ground monitoring stations to comprehensively estimate $PM_{2.5}$ concentration in various regions of China [28]. The $PM_{2.5}$ concentration data from NASA satellite data and ground monitoring stations show the concentration changes from 2000 to 2017 in Figure 3. In addition, the Environmental Bulletin of Guangdong Province has reported the environmental quality of Guangdong Province since 2012, as shown in Figure 3 [2,29–33], based on comprehensive statistics data from multiple monitoring points. In the past, due to the lack of local monitoring data, $PM_{2.5}$ in Guangdong Province was approximately 17 $\mu g/m^3$ in 2000. As a result of rapid economic growth, energy consumption gradually increased, and the concentration of $PM_{2.5}$ rose, reaching 35 $\mu g/m^3$ by 2008. Then, Guangdong Province began vigorously promoting the establishment of reduction technology, and $PM_{2.5}$ concentrations began to decrease. However, according to the Environmental Statistics Bulletin data of Guangdong Province, $PM_{2.5}$ decreased from 47 $\mu g/m^3$ in 2012 to about 31 $\mu g/m^3$ in 2017 [34]. With regard to the concentration of $PM_{2.5}$, there is a gap between the two statistical sources due to the different layouts of the pollutant monitoring points. Finally, through comparison, in order to reflect the characteristics of Guangdong monitoring stations, the statistical data of Guangdong Province in 2015 are adopted and 34 $\mu g/m^3$ is used as the reference value of the base year in the model.

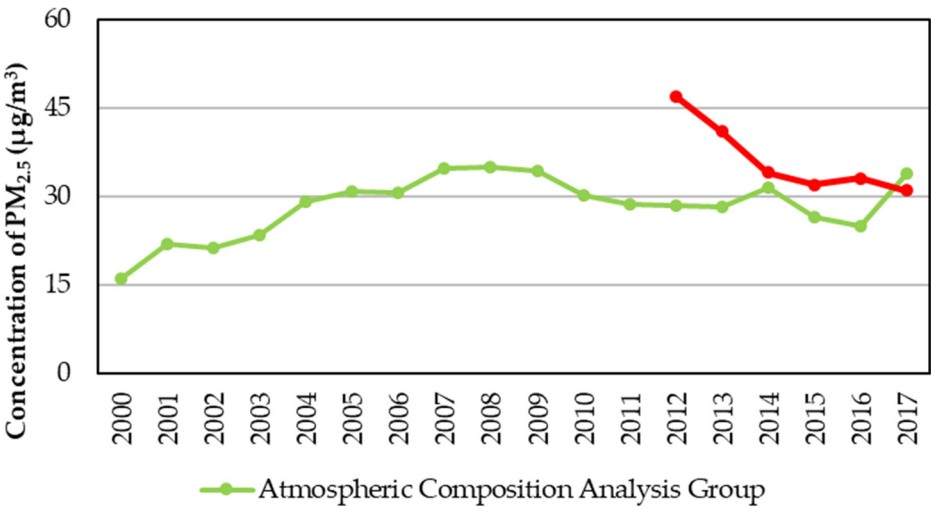

**Figure 3.** PM$_{2.5}$ concentration in Guangdong.

### 2.2.3. Population and Labor Force

The labor force and population of Guangdong Province have been growing rapidly. Because Guangdong Province is one of the most developed provinces in China, it holds strong attraction for migrant workers. From 2005 to 2017, the average annual population growth rate reached 1.6%, and the average annual growth rate of the labor force reached 1.97%, thus exceeding the population growth rate [27]. The labor force has greatly contributed to economic growth, as shown in Figure 4. As a result of this growth in population, the medical expenditure and welfare loss caused by PM$_{2.5}$ will increase together.

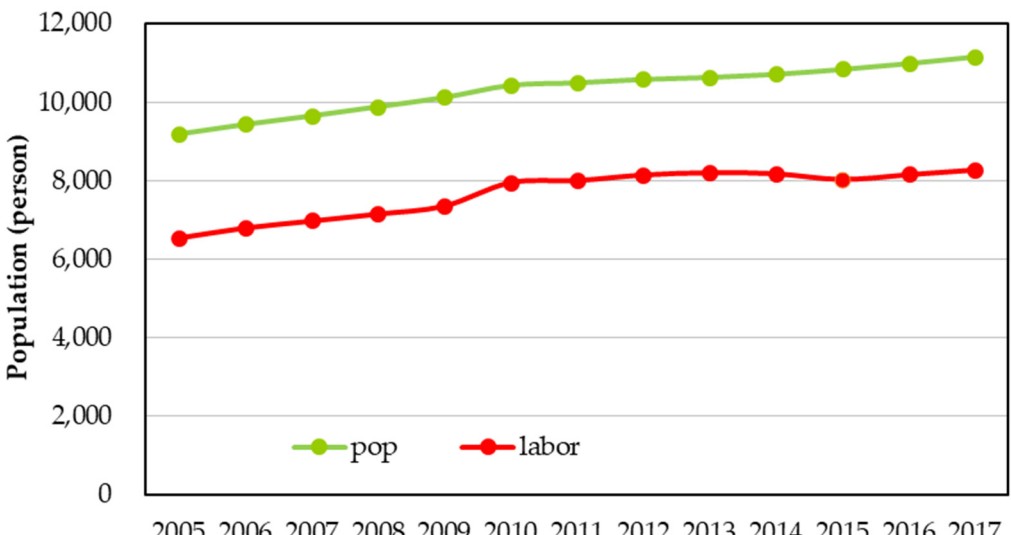

**Figure 4.** Population and labor force (15–65 years of age) growth rate from 2005–2017 in Guangdong.

### 3. Scenario Description and Setting

The variables of the model settings are described in Table 2. Three scenarios were established: WOC (without air control), AIC (within air control equipment), and BLK (within enhanced pollutant treatment equipment for blue sky). The WOC scenario is to achieve China's INDC target by 2035, but the pollutant control strategy is consistent with that of 2015. The energy consumption results is shown in Table A2. Based on the WOC scenario, the AIC scenario strengthens the pollutant control strategy and achieves the target of PM$_{2.5}$ concentration below 30 µg/m$^3$ by 2035. In the BLK scenario, the PM$_{2.5}$

concentration in Guangdong Province will be reduced to WHO level II 25 $\mu g/m^3$ by 2035 with a stricter pollution control strategy in the GAINS model. Under the three scenarios, the average annual investment growth rate of Guangdong Province is set at 8% for 2010–2015, 7% for 2015–2020, and 6.5% for 2020–2035 in the CGE model. The three scenarios set the same carbon constraint target, with constant investment drivers and consistent energy transformation levels of corresponding subsectors. The WOC scenario is a no additional air pollution control scenario, and the level of air pollution control remains at the level of 2015, whereas the additional emissions caused by energy use are not effectively controlled. AIC applies many kinds of air pollution terminal control technologies in its GAINS model, and $PM_{2.5}$ emissions will change under the control scenario. The BLK scenario is a stricter air pollution control strategy and more technology will be installed, and $PM_{2.5}$ concentration will by further reduced.

**Table 2.** Scenario settings.

| Scenario Name | CGE Model Parameter Setting: GDP Growth Rate | CGE Parameter Setting | GAINS Parameter Setting Description: Industry | GAINS Parameter Setting Description: Transport |
|---|---|---|---|---|
| WOC | | | Considering health impact of the labor force. Until 2035, the air control strategy is same as in 2015. (1) The existing coal-fired boiler adopts electric dust removal equipment which maintains the status quo. (2) the newly added boiler is also the same as the existing boiler using the same electric dust removal equipment | (1) For the different vehicles in road transport, the EURO 3 level standard on light duty spark ignition for road vehicles is adopted. (2) Consider more electric vehicles being adopted at low speeds of 5%. |
| AIC | In the years of 2015 to 2020, the GDP growth rate is set as 7%; in the years of 2020 to 2030 and 2030 to 2035, the GDP growth rate is set as 6% and 5.5% according to the economic development plan. | Considering the carbon emission constraint target policy for Guangdong: Toward 2030 achieving INDC target, $CO_2$ intensity reduced by 60–65% | Air control strategy is moderate. (1) The penetration rate of electric dust removal equipment for existing coal-fired boilers has reached more than 80%, and the removal rate has reached more than 96%. (2) The newly-added boiler adopts high-efficiency electric dust removal equipment, and the dust removal efficiency has reached more than 99%. | (1) By 2035, using Stage 2 control technology on heavy duty vehicles, and executing the EURO 4 level standard on light duty spark ignition road vehicles. (2) Consider that more electric vehicles are adopted at medium speed of 10%. |
| BLK | | | Air control strategy is strict. (1) $PM_{2.5}$ concentration reaches WHO level II (25 $\mu g/m^3$). (2) By 2035, industry $PM_{2.5}$ removal efficiency is 99%. | (1) Using Stage 3 pollution control technology on heavy duty vehicles. (2) Implement EURO 5 standard on light duty for spark ignition road vehicles. (3) Considering the adoption of more electric vehicles at high speed of 20%. |

## 4. Results and Analysis

### 4.1. Quality of Air Pollutants Based on GAINS Model

As shown in Figure 5, in the WOC scenario, the $PM_{2.5}$ concentration simulated by the GAINS model will be 32.61 $\mu g/m^3$ in 2025 and will drop to 32.60 and 30.63 $\mu g/m^3$ in 2030 and 2035, respectively. In the AIC scenario, it will decline to 29.28 $\mu g/m^3$ by 2025, 28.74 $\mu g/m^3$ by 2030, and 28.54 $\mu g/m^3$ by 2035. In the BLK scenario, it will decline to 27.60 $\mu g/m^3$ by 2025, and 26.48 and 25.51 $\mu g/m^3$ by 2030 and 2035, respectively, through

the adoption of more stringent emission control of industrial terminal equipment and oil substitution upgrades in the transportation sector.

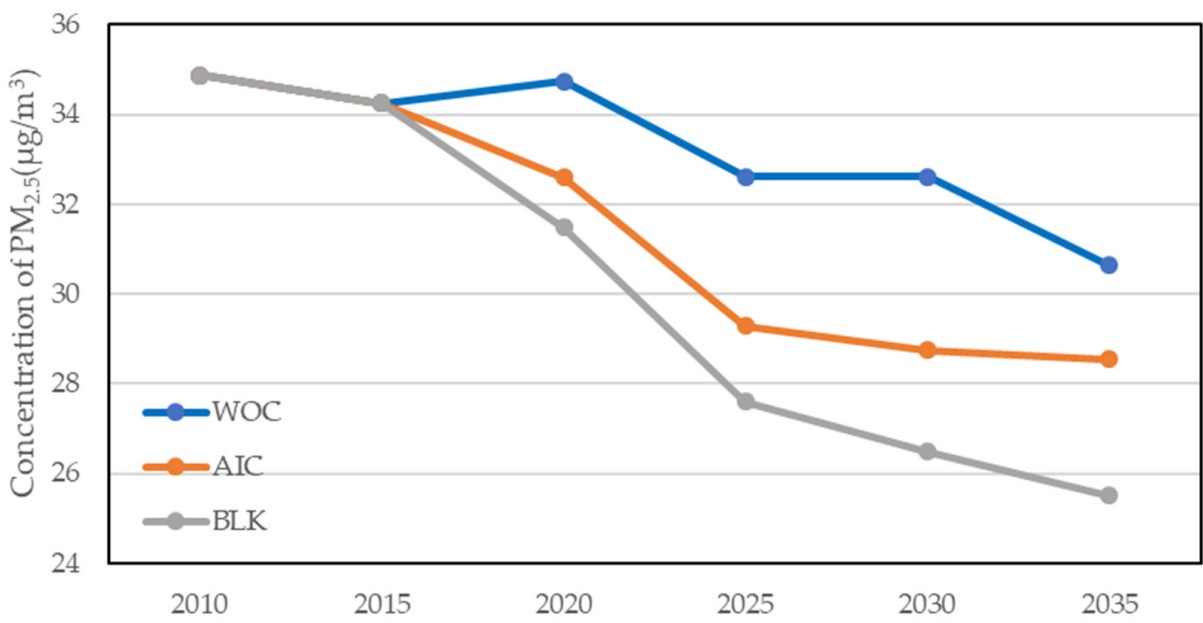

**Figure 5.** PM$_{2.5}$ concentration from 2010 to 2035.

### 4.2. Emission Sector Analysis

The main emission sectors of PM$_{2.5}$ are cement, electric power, and residential consumption. After implementation of the air pollutant control strategy, the main emission sectors are electric power and residential sector. As shown in Figure 6, in the WOC scenario, the sector with the largest PM$_{2.5}$ emissions in 2025 is the cement sector (69.9%), followed by the power sector above 50 MWth (12.5%). After 2035, the coal-fired generating units in Guangdong Province are expected to be eliminated and upgraded. The stock of coal-fired generating units has dropped by 100%, and the total amount of power sector has dropped by 3.5%. Compared with the WOC scenario, the BLK and AIC scenarios have implemented more strict air control strategy, and the total volume of PM$_{2.5}$ emission in 2025 will drop by 46.9% and 63.9%, respectively. The cement sector declines the most, decreasing by 63.1% and 81.4%. In 2035, PM$_{2.5}$ emissions fall by 50% and 79.9%, respectively. The cement sector is still the sector with the largest decline, falling by 64.3% and 91.5%.

The PM$_{2.5}$ emissions in the transportation sector mainly include those of eight types of transportation: heavy duty vehicles, buses, light duty vehicles, cars, motorcycles, mopeds, railways, and inland waterways. The PM$_{2.5}$ emissions in the transportation sector account for about 6.6% of total PM$_{2.5}$ emissions in the whole society. Due to the mandatory installation of PM$_{2.5}$ control measures by the industrial sector, the industrial PM$_{2.5}$ emissions decrease quickly. Thus, the proportion of PM$_{2.5}$ emissions in the transportation sector increases gradually. By 2035, the PM$_{2.5}$ emission of the transportation sector will account for 5.8% under the WOC scenario, but the PM$_{2.5}$ emission proportion rises to 11.7% under the AIC scenario, then under the BLK scenario the proportion of emissions further increases to 29.1%. The emission reduction in PM$_{2.5}$ in the transportation industry is worthy of significant efforts and needs more attention.

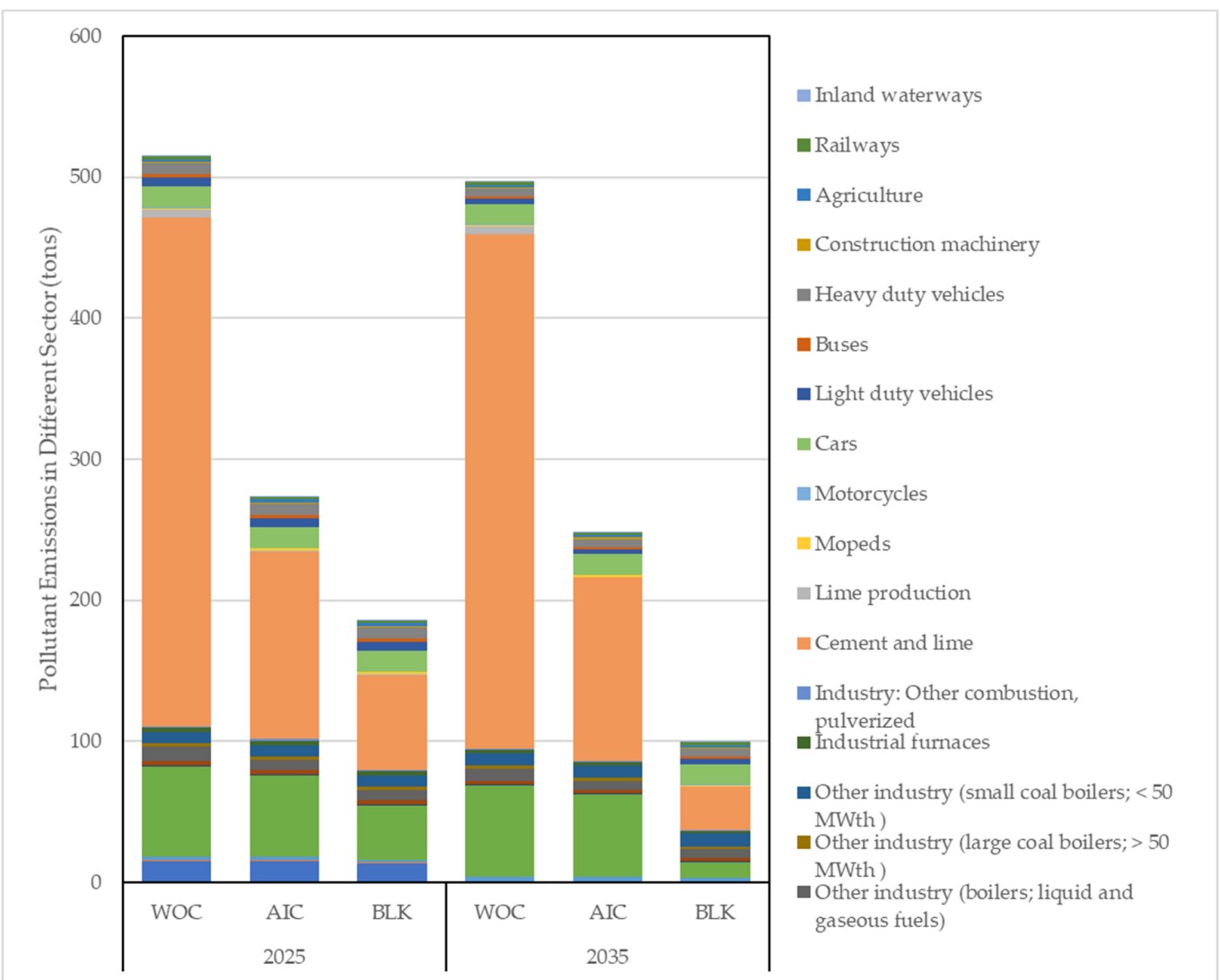

**Figure 6.** The PM$_{2.5}$ emission contributions from different sectors.

*4.3. Health Impact*

Many studies confirm that long-term exposure to high PM$_{2.5}$ pollution increases the incidence rate and mortality from respiratory, cardiovascular, and cerebrovascular diseases. By 2025 and 2035, the INDC of the air pollutant control level in 2015 will be adopted in Guangdong Province, as shown in Figure 7. The incidence rate of the WOC scenario is about 240 cases/10,000 persons. The incidence rate of the AIC and BLK scenarios is 175 cases/10,000 persons and 160 cases/10,000 persons in 2025, respectively, and 150 cases/10,000 persons and 140 cases/10,000 persons in 2035, respectively. In addition, premature deaths will number 71,690, 70,000, and 40,000 in 2035 for the three scenarios, respectively. As shown in Figure 8, the resulting loss of working time is 0.4–0.7 days per person; a strict air pollution control strategy can reduce the health impact.

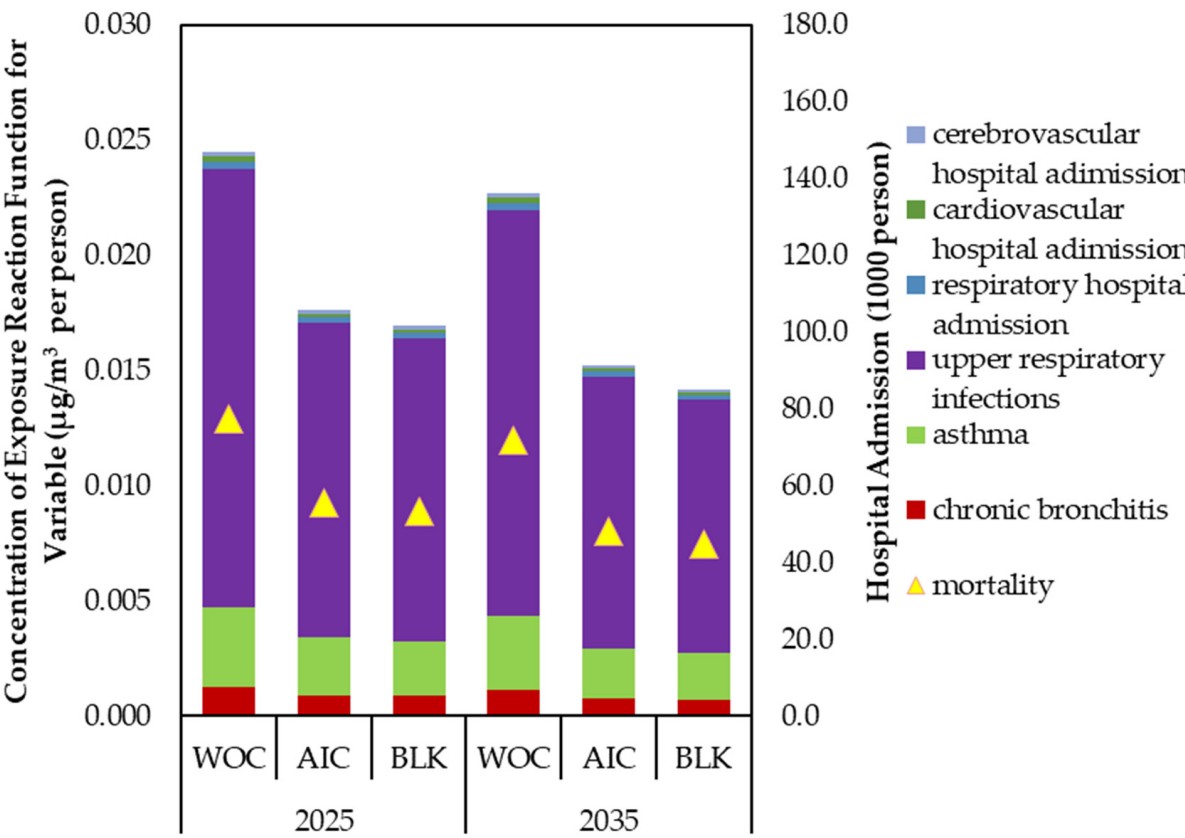

**Figure 7.** Hospital admissions and mortality cases caused by PM$_{2.5}$ concentration.

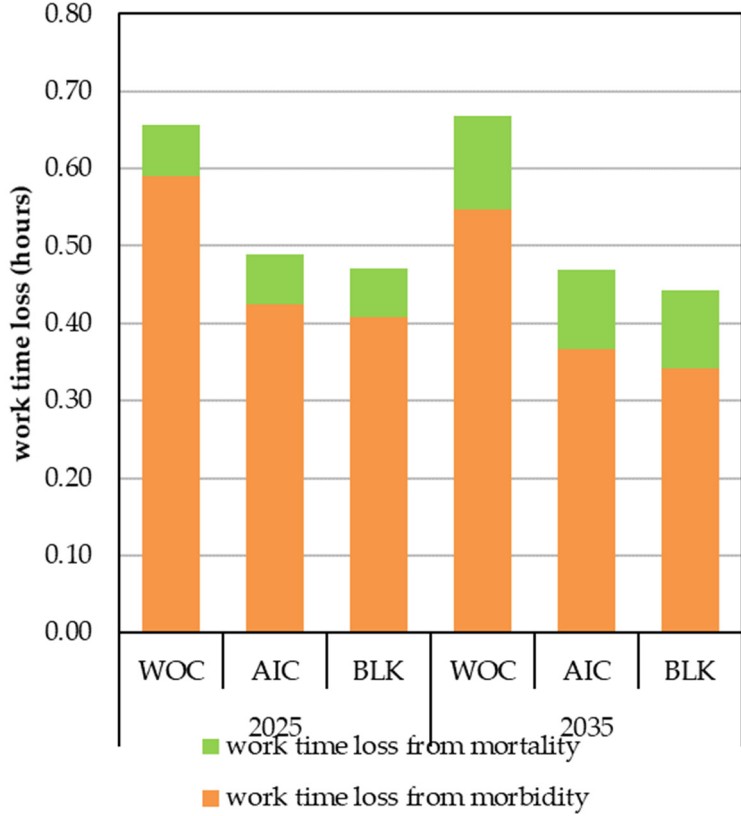

**Figure 8.** Work loss days caused by PM$_{2.5}$ concentration.

### 4.4. Macro-Economic Impact

PM$_{2.5}$ pollution leads to an increase in the number of patients, and the additional medical expenses and loss of working hours caused by related diseases will impact the macro-economy. PM$_{2.5}$ pollution increases the number of cases of related diseases, which will lead to additional medical expenses, thus affecting disposable income and people's welfare. At the same time, PM$_{2.5}$ pollution causes related diseases, which will cause the working population to spend more time in medical treatment each year, resulting in work loss time, as reflected in the decline of the labor force level in the CGE model. Compared with the WOC scenario, Table 3 shows that in the AIC scenario, the GDP and welfare gains in 2035 are 0.05% and 0.28%, respectively. In the BLK scenario, sector output will also be affected, especially in labor-intensive sectors, such as agriculture, food, textiles, and service industries, which will promote output gains. By 2035, the gains in GDP and welfare will be 0.11% and 0.36%, respectively. Adopting more stringent air pollution control strategies will increase GDP and welfare.

**Table 3.** GDP and welfare loss.

|  | GDP Gains (%) | | | Welfare Gains (%) | | |
| --- | --- | --- | --- | --- | --- | --- |
|  | **Lower** | **Medium** | **Upper** | **Lower** | **Medium** | **Upper** |
| AIC | 0.2 | 0.05 | 1.28 | 0.2 | 0.28 | 0.37 |
| BLK | 0.35 | 0.11 | 0.36 | 0.26 | 0.36 | 0.46 |

As shown in Figure 9, compared with the WOC scenario, the BLK scenario has differing impacts by sector. Among the most impacted sectors are labor-intensive industries such as agriculture, food production, other manufacturing industries, and textiles, whereas the impact on pipeline transportation and waste disposal industries is negative. It is found that the transportation sector has little economic impact due to the current policy on pollutant emission reduction in the transportation sector is not as strict compared with the industry. Although different degrees of air control measures are adopted in the three scenarios, the energy transformation of the transportation sector has a different path to that of other sectors, mainly because the electric vehicles are guided to replace traditional fuel vehicles at varying speeds, resulting in the promotion effect of industrial added value and the optimization of the energy structure. The added value of the transportation sectors, such as road (0.105%), railway (0.137%), waterway (0.078%), and aviation (0.092%), is increased.

### 4.5. Cost-Benefit Analysis

The GAINS model calculates the cost of different air pollution control strategies (in millions of euros/year). As shown in Table 4, under the WOC scenario, the cost of air pollution control increases from 375 million euro in 2010 to 400 million euro in 2035, whereas under the AIC and BLK scenarios, it increases to 464 million euro and 639 million euro, increasing by 16% and 60%, respectively.

**Table 4.** Air pollution control cost.

|  | **2015** | **2020** | **2025** | **2030** | **2035** |
| --- | --- | --- | --- | --- | --- |
| WOC | 394.97 | 442.76 | 441.31 | 426.15 | 400.89 |
| AIC | 394.97 | 503.38 | 501.73 | 492.16 | 464.69 |
| BLK | 394.97 | 523.01 | 589.72 | 616.56 | 639.08 |

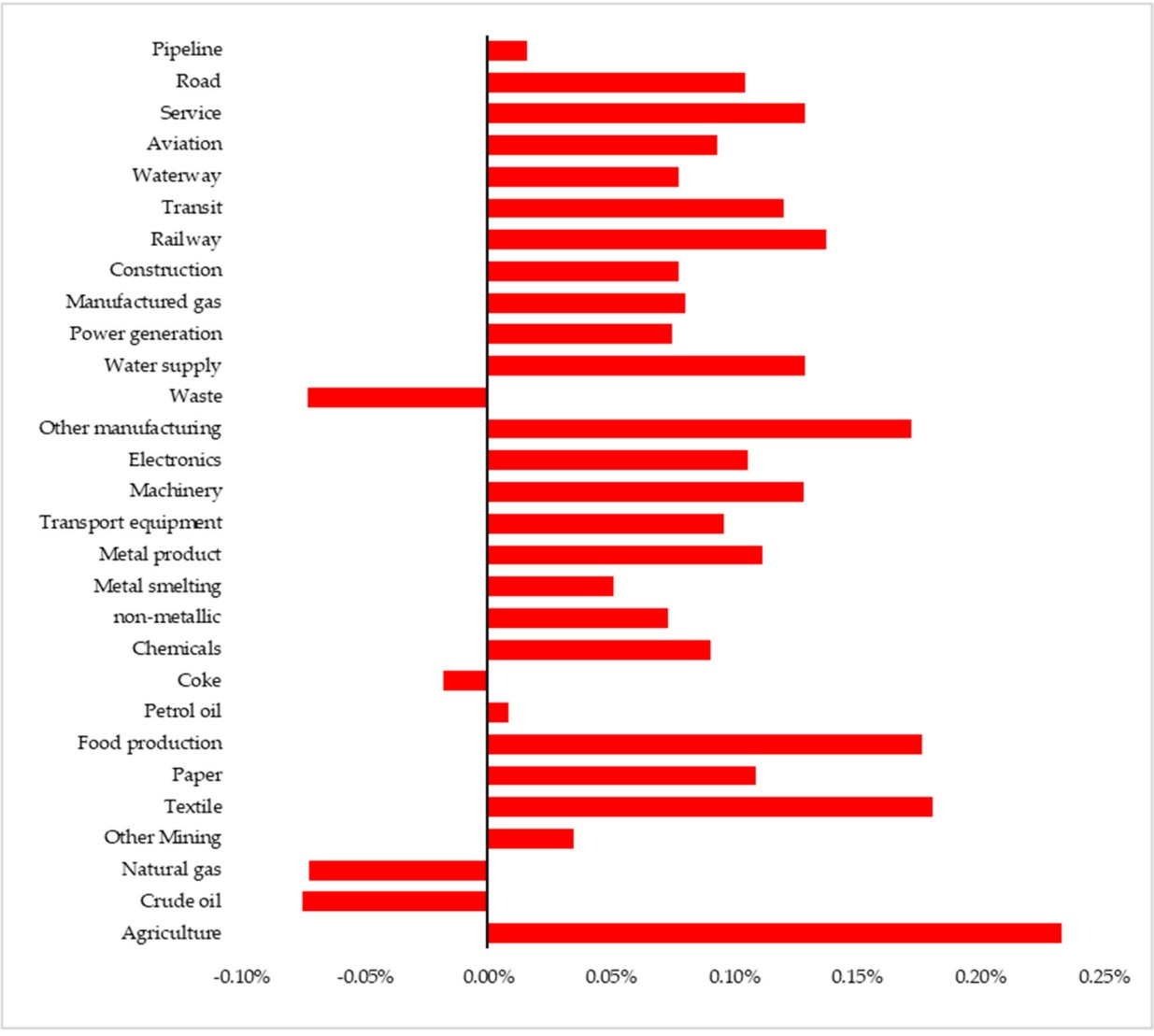

**Figure 9.** Value-added change in the BLK scenario compared with the WOC scenario.

By comparison, increasing the cost of pollutant treatment has also effectively reduced the concentration of pollutants, and the decline in the concentration of pollutants has also further reduced the medical costs of the exposed population and the decline in the quality of labor. The improvement in the health level is further input into the CGE model, resulting in GDP growth compared to the WOC scenario; the gains of AIC and BLK in terms of GDP are 0.05% and 0.11%, respectively.

As shown in Figure 10, from 2025 to 2035, compared with the benchmark WOC scenario, the penetration ratio of pollutant treatment facilities in AIC and BLK scenarios increases, and the cost of pollutant treatment is greater than the benefit in 2025, and gradually reverses by 2035. Compared with WOC scenario, the ratios of benefit/cost of the AIC and BLK scenarios are 0.76 and 1.19, respectively, in 2035. The benefit resulting from the increase in labor quality is greater than the treatment cost, leading to a net welfare benefit to the whole society. Thus, for different locations, it is necessary to comprehensively consider the impact of improving air quality.

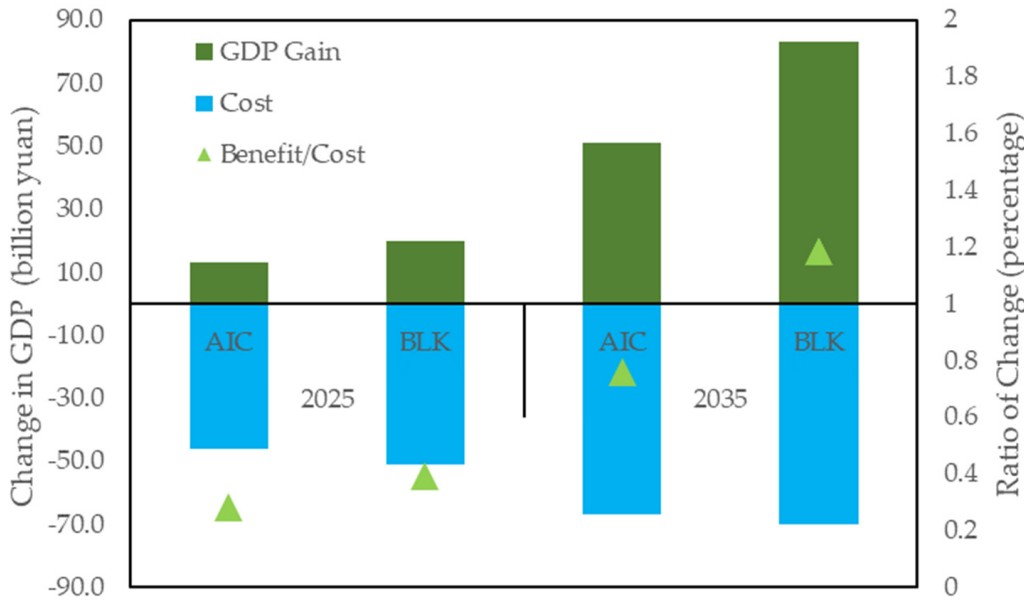

**Figure 10.** Expenditure and cost in 2025 and 2035.

### 4.6. Uncertainty Analysis

One of the uncertainties of this study is how to further refine the benefits of pollutant emission reduction on the heath of the labor force. The results from estimating the total number of premature deaths due to PM$_{2.5}$ pollution in Guangdong Province in 2010 and 2035 using different exposure response functions (ERFs) are presented. The ERF determines all-cause mortality estimated using a linear exposure response function [25], which is based on the adjusted Cox proportional risk model. Non-linear (non-linear IER) refers to the mortality rate of five diseases (ischemic heart disease, stroke, chronic pulmonary obstructive pneumonia, lung cancer, and acute lower respiratory tract infection) caused by PM$_{2.5}$ calculated according to the concentration RR value control table of the GBD database. The mortality rate of the five diseases is estimated by a nonlinear GEMM (Global Exposure Mortality Model) equation using the GEMM exposure effect function [35].

Although mortality estimates vary between exposure effect functions, it is clear that under the emission reduction scenarios (AIC and BLK), the mortality of Guangdong province in AIC and BLK scenario is lower compared with the situation of no air pollution control (WOC) (Figure 11). For example, the premature death toll due to PM$_{2.5}$ air pollution in Guangdong Province estimated by the GEMM equation is highest, followed by the total deaths estimated by the linear equation and the nonlinear function. Among the three exposure effect functions, the uncertainty of the linear function is highest for the total number of premature deaths, whereas the uncertainty of the GEMM equation is lowest.

When taking emissions reduction measures, in 2035 in the AIC scenario, based on the nonlinear function, the total number of premature deaths in Guangdong Province is expected to decrease to 87,000 (38,000–124,000), which is estimated to be 112,000 (76,000–144,000) based on the GEMM function. Based on the linear equation, the total number of deaths is expected to decrease to 76,000 (27,000–131,000); in the BLK scenario, the total number of premature deaths in Guangdong Province is expected to decline to 79,000 (3.5–11.5 million) based on the nonlinear IER function, 104,000 (7.0–13.4 million) based on the GEMM function, and 64,000 (2.2–11.1 million) based on the linear IER equation.

In 2035, compared to the AIC scenario, in the two policy scenarios, the number of premature deaths in the BLK scenario decreases more significantly (Figure 12). Among the three exposure effect functions, in the AIC scenario, the linear effect function shows the highest decline rate (9.6%), followed by the nonlinear equation (5.0%), with GEMM being the lowest (4.7%). Similarly, in the BLK scenario, the linear effect function showed

the highest decline rate (23.7%), followed by the nonlinear equation (13.1%), and GEMM was lowest (11.9%). A comparison of the errors of the two methods indicates this result is acceptable.

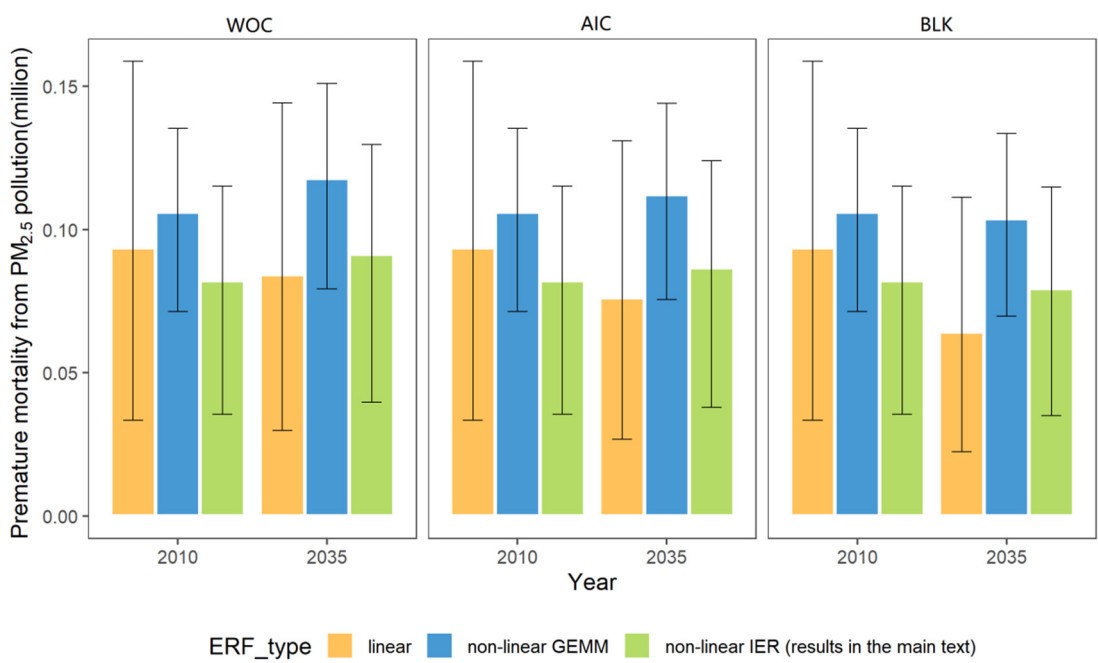

**Figure 11.** Premature deaths due to $PM_{2.5}$ pollution in Guangdong under different scenarios.

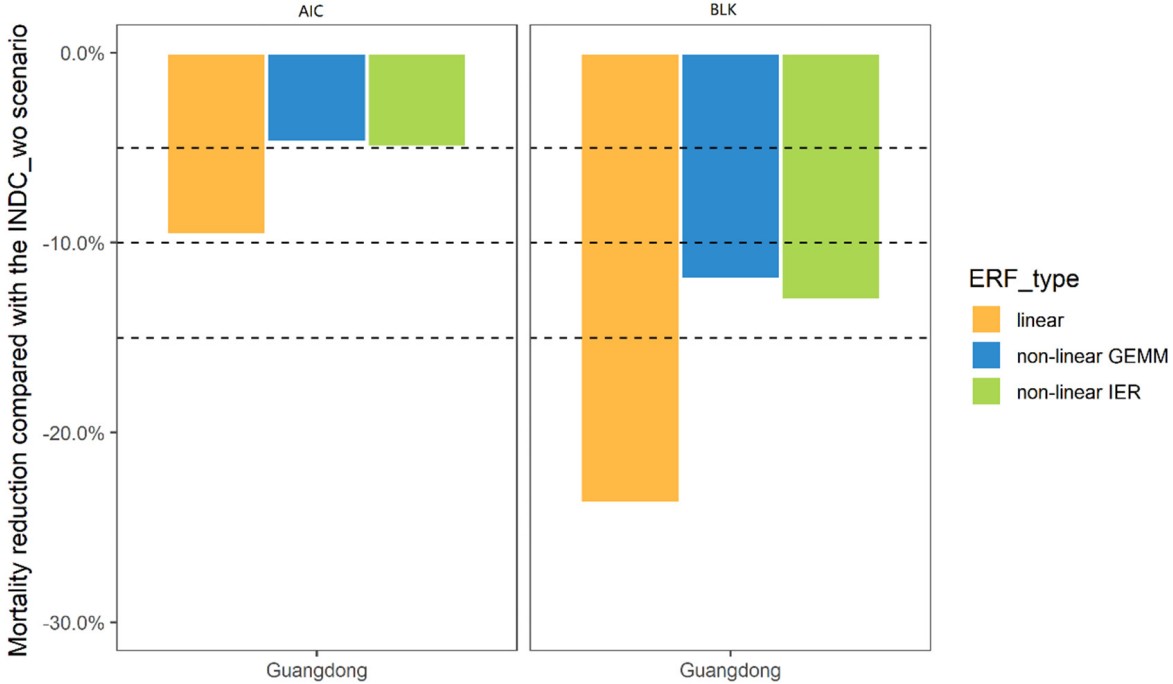

**Figure 12.** Mortality reduction rate of policy scenarios in Guangdong Province in 2035 (compared with the WOC scenario).

This study analyzed the sensitivity of the ERFs to health and economic variables. The results show a −17% to 17% morbidity interval, a −93% to 100% chronic disease mortality interval, a −48% to 37% expenditure interval, and a −27% to 29% work time loss interval, which are equivalent to a 2030 ERF 95% confidence interval (CI). This shows that chronic mortality caused by air pollution is very sensitive to the ERF. However, only 10% of lost

working time stems from mortality, so the loss of working time differs from the economic indicators. Table 3 shows that the lower and upper limits of GDP and welfare losses are between −26% and 32%, equivalent to a 95% CI of ERF.

## 5. Discussion

Previous research [5–7] has considered the positive externalities resulting from the treatment of pollutants and reached the conclusion that the impact on the local regional economy is positive. Other scholars [12,14] did not consider the macro-economy of the system and concluded that pollution control resulted in large investment costs, or air pollution resulted in economic losses of 1–3%. This study described the impact of the whole life cycle of energy consumption to support economic growth, while producing air pollutants, on the environment and health. The analysis of the cost and economic benefits of controlling air pollutants will improve the cost effectiveness of the government's action of formulating emission reduction policies. This can persuade public finance to make significant efforts to control air pollution, which is a public product, and that investment in pollutant treatment can result in benefits. For the production and transportation activities of the industrial and transportation sectors, in particular, if more mandatory pollutant treatment measures and improvements in emission reduction standards can be implemented, it will not necessarily have a negative impact on the activities of the industry. On the contrary, it will result in a greater increase in added value over time. The long-term reversal of cost-effectiveness also shows that the generation of environmental protection benefits requires sustained investment and sustained financial support, so as to cause sustainable benefits from the aspects of health effect and improvement in labor quality.

Pollutant treatment can not only reduce the health risk caused by long-term exposure to high pollutant concentrations and improve the health level, but also improve the quality of the labor force. In addition to unemployment time, air pollution hinders economic development by reducing labor productivity. If the impact on labor productivity was considered, the impact on the economy would have been greater than what is shown in our results. This study also underestimates the impact of $PM_{2.5}$ pollution from this perspective. Sustained environmental protection policies and capital investment will result in long-term benefits to Guangdong, thus promoting the construction of a world-class urban agglomeration, and an environment suitable for living, business and tourism, and a healthy life.

The energy consumption in each scenario in this study fully considers the problems of industrial transfer and energy structure transformation in the economic development of Guangdong Province. These will reduce the fossil energy consumption of Guangdong Province and improve the production efficiency of the whole society. Through the combination of these energy scenarios and the GAINS model, we can better evaluate the future environmental impact assessment of Guangdong Province.

In addition, this study only examined the $PM_{2.5}$ emissions from energy production and consumption in Guangdong Province. The concentration of $PM_{2.5}$ in other adjacent provinces considered in this study remained at the level of 2015, regardless of their increase or decrease in $PM_{2.5}$. As a result of further strengthening of air pollution control in each adjacent province, the background $PM_{2.5}$ concentration in Guangdong Province will be further reduced in the future.

## 6. Conclusions

The health impacts of $PM_{2.5}$ pollution in Guangdong Province from 2015 to 2035 were evaluated. The following can be drawn:

(1) Compared with the WOC scenario, the BLK and AIC scenarios implemented more strict air control strategies, and the total volume of $PM_{2.5}$ emission in 2025 will drop by 46.9% and 63.9%, respectively. The cement sector declines the most, decreasing by 63.1% and 81.4%, respectively. In 2035, $PM_{2.5}$ emissions fall by 50% and 79.9%,

respectively. The cement sector is still the sector with the largest decline, falling by 64.3% and 91.5%, respectively.

(2)  The incidence rate of the WOC scenario is about 240 cases/10,000 persons. The incidence rate of the AIC and BLK scenarios is 150 cases/10,000 persons and 140 cases/10,000 persons in 2035, respectively. In addition, premature deaths will be 70,000 and 40,000 in 2035 for the two scenarios, respectively. Due to the control of pollutants, the early death rate decreases by 33.0% and 37.5%, respectively, and the health effect improve.

(3)  Furthermore, improved labor force quality induced by better air conditions promotes increased added value in labor-intensive industries such as agriculture (0.233%), other manufacturing (0.172%), textiles (0.181%), food (0.176%), railway transport (0.137%), and services (0.129%). The added value declines in the waste ($-0.073\%$), natural gas ($-0.076\%$), and crude oil sectors ($-0.072\%$) because more $PM_{2.5}$ treatment equipment requires greater investment.

The results show that strengthening air pollution control can reduce $PM_{2.5}$ pollution, reduce the impact on health, and improve the health quality of the labor force, including reductions in employment time and labor expenditure, thus promoting overall economic growth. Considering the cost of the control technology, the improvement in $PM_{2.5}$ pollution in 2035 will result in a net benefit of 0.11% of GDP.

Although treatment of pollutants will lead to additional costs, the health impact analysis shows that it will also result in positive health effects, increase the quality of the labor force, and reduce disease expenditure. Thus, investment in air control technology in Guangdong Province appears to be favorable.

This case study helps to support the government's environmental protection investment and the implementation of environmental protection policies. Compared with the impact of pollutants on human health in the WOC scenario, the cost of the BLK scenario is still less than the benefit derived from the reduction in pollutants.

The net benefit is higher in the food, textile, and service sectors. Policy makers should take these differences into account and make appropriate policies for different sectors. Because electric vehicles can not only promote the emission reduction in the transportation sector, but also have a certain economic effect on promoting industrial development and employment, the application of such measures in environmental governance and technological renewal should be strengthened. However, attention should also be paid to the pollution after the service life of the battery expires. Air pollution concentration depends not only on the control of front-end energy consumption, but also on the control level of pollutant control technology. It needs the full cooperation between the government departments that promote industrial production and environmental governance. This requires government departments to work together to formulate policies to reduce air pollution.

**Author Contributions:** Methodology, H.D. and T.M.; Project administration, S.R.; Software, H.D.; Writing—original draft, S.R.; Writing—review & editing, P.W. and D.Z. All authors have read and agreed to the published version of the manuscript.

**Funding:** This research was funded by National Natural Science Foundation of China (71603248) and Science and Technology Planning Project of Guangdong Province (2017A050501060).

**Institutional Review Board Statement:** Not applicable.

**Informed Consent Statement:** Not applicable.

**Data Availability Statement:** Not applicable.

**Conflicts of Interest:** The authors declare no conflict of interest.

## Appendix A

**Table A1.** Exposure response functions.

| Impact Category | ERFs | C.I. (95%) Low | C.I. (95%) High |
|---|---|---|---|
| Chronic mortality | 0.40% | 0.03% | 0.80% |
| Respiratory hospital admissions | $1.17 \times 10^{-5}$ | $6.38 \times 10^{-6}$ | $1.72 \times 10^{-5}$ |
| Cerebrovascular hospital admission | $8.40 \times 10^{-6}$ | $6.47 \times 10^{-7}$ | $1.16 \times 10^{-5}$ |
| Cardiovascular hospital admissions | $7.23 \times 10^{-6}$ | $3.62 \times 10^{-6}$ | $1.09 \times 10^{-5}$ |
| Chronic bronchitis | $4.42 \times 10^{-5}$ | $-1.82 \times 10^{-6}$ | $9.02 \times 10^{-5}$ |
| Asthma attacks | $1.22 \times 10^{-4}$ | $4.33 \times 10^{-5}$ | $12.08 \times 10^{-4}$ |
| Respiratory symptoms days | $2.50 \times 10^{-2}$ | $2.17 \times 10^{-1}$ | $4.05 \times 10^{-1}$ |
| work loss day | $2.07 \times 10^{-2}$ | $1.76 \times 10^{-2}$ | $2.38 \times 10^{-2}$ |

**Table A2.** Energy consumption for each sector in Guangdong (Mtce).

| Sectors | 2020 | 2025 | 2030 | 2035 |
|---|---|---|---|---|
| Agriculture | 5.59 | 6.56 | 6.09 | 5.65 |
| Power generation | 48.87 | 48.91 | 46.50 | 44.22 |
| Petroleum mining | 0.06 | 0.01 | 0.00 | 0.00 |
| Natural gas mining | 2.95 | 3.03 | 1.63 | 0.88 |
| Other mining | 1.67 | 1.97 | 2.04 | 2.11 |
| Food manufacturing | 5.34 | 5.97 | 5.72 | 5.48 |
| Textile | 5.04 | 5.27 | 4.42 | 3.71 |
| Wood processing | 1.69 | 1.76 | 1.69 | 1.64 |
| Papermaking | 9.44 | 10.28 | 10.91 | 11.59 |
| Other manufacturing | 6.71 | 7.12 | 6.65 | 6.22 |
| Oil refining | 12.09 | 14.44 | 16.29 | 18.38 |
| Coking | 11.53 | 8.83 | 2.69 | 0.82 |
| Chemical industry | 27.93 | 33.32 | 37.78 | 42.85 |
| Cement | 11.92 | 12.44 | 12.67 | 12.90 |
| Other non-metallic manufacturing industry | 10.59 | 11.78 | 10.13 | 8.71 |
| Glass manufacturing | 4.18 | 4.50 | 3.70 | 3.04 |
| Ceramic manufacturing | 3.47 | 3.97 | 3.65 | 3.37 |
| Steel | 74.75 | 89.20 | 106.91 | 128.13 |
| Non-ferrous metal smelting | 7.43 | 9.04 | 9.86 | 10.76 |
| Metalware | 8.95 | 10.03 | 10.22 | 10.41 |
| Mechanical manufacturing | 11.09 | 12.27 | 12.78 | 13.30 |
| Electronic equipment manufacturing | 16.51 | 15.57 | 13.79 | 12.21 |
| Gas production and supply | 1.01 | 1.12 | 1.01 | 0.92 |
| Water production and supply | 2.00 | 2.29 | 2.52 | 2.77 |
| Construction | 3.88 | 4.43 | 4.54 | 4.65 |
| Road transport | 1.81 | 2.17 | 2.22 | 2.28 |
| Railway transportation | 9.38 | 11.58 | 11.71 | 11.84 |
| Urban public transport | 7.39 | 10.48 | 10.66 | 10.84 |
| Water transport | 10.64 | 12.59 | 11.45 | 10.41 |
| Air transport | 11.17 | 13.40 | 15.64 | 18.24 |
| Other transportation | 0.07 | 0.08 | 0.07 | 0.06 |
| Service industry | 18.06 | 22.90 | 25.35 | 28.05 |
| Total | 353.20 | 397.31 | 411.30 | 436.42 |

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
