# Peer review of "Health and Economic Impact Assessment of Transport and Industry PM2.5 Control Policy in Guangdong Province"

_sustainability, doi:10.3390/su132313049_

Round 1

Reviewer 1 Report

It was a bit difficult to read this article, because of the uploaded version where it is seen all corrections. Meanwhile, it is very clear what was corrected and improved. The article was quite a lot improved: figures constructed correctly with axes, units, etc., the description is better, English language and style are fine. 

Author Response

Thank you for your guidance. We have further sorted out the full text to ensure that the icons, units, subscripts and subscripts, references, sentences, etc. are more in line with the writing standards of scientific and technological papers.

Reviewer 2 Report

The review on the manuscript in journal Sustainability entitled „Health and economic impact assessment of Transport and Industry PM2.5 control policy in Guangdong Province“.

The article analyzes the additional damages related to PM2.5 pollution (additional medical expences, mortality, etc.) in Guangdong Province under different scenarios and ways to reduce them.

Broad comments

The description of the models used in the analysis is practically very general. The relationships and calculation equations underlying the models are completely missing from the article. The assumptions and limitations and possible uncertainties of the models used have not been analyzed in detail. It also does not make it possible to evaluate the results obtained.

The technical implementation of the article does not meet the standards of academic writing.

Academic writing should be objective. If it is subjective or emotional, it will lose persuasiveness and may be regarded as relying on emotion rather than building a reasonable argument based on evidence. The language or informal writing should therefore be impersonal, and should not include personal pronouns. For most subject areas the writing is expected to be objective. For this the first person (I, we, me, my, etc.) should be avoided. In this article on line 13 is written “ we construct", line 16 „We analyze“, line 146 „we identify“, line 153 „we build“, line 162 „We then estimate“, line 205 „we can use“, line 237 „we use“, line 238 „we compare“, line 250 „we obtain“, line 311 „We set“, line 510 „we also consider“ and „we find“, line 519 „We evaluate“, line 527 „we consider“, line 528 „we also need“ and line 541 „we find“. Eliminating personal pronouns from writing is highly recommend.

Specific comments

The same style should be used throughout the article, either PM2.5 (lines 3, 11, 13, 29, 30, 57, 63, 71, 77, 83, 110, 137, 146, 157, 158, 160, 162, 203, 236, 239, 242,243, 245, 247, 252, 264, 269, 304, 327, 328, 331, 334, 440, 538 and 540) or PM 2.5 (lines 226, 248, 274, 277, 280, 292, 284, 287, 295, 316, 317, 344, 346, 351, 357, 360, 362, 363, 364, 365, 370, 372, 385, 388, 390, 392, 394, 434, 448, 465, 512, 513, 517, 519 and 542).

Formulas for chemical substances must be formulated correctly using subscripts – lines 41, 42 „CO2“, line 74 „SO2“, line 75 „NO2“, line 82 „NO2“, line 91 „O3“, line 203 „SO2, NOx“, line 204 „NH3“.

Units need to be formatted correctly using superscripts– lines 63, 65, 71, 247, 278, 280, 285, 317, 331, 335, 336, 337, 338 and 339.

References to sources must be in square brackets and must not be a superscript – lines 155, 156, 157, 175, 235, 240, 259, 265, 274, 276, 285, 302, 437 and 443.

All Figures and Tables must be referenced in the text. All Figures and Tables should be inserted into the main text close to their first citation and must be numbered following their number of appearance (Figure 1, Figure 2, Table 1, etc.). There is complete confusion in this article.

When referring to Figures, the same style should be used in the article – line 303 „figure 4“, line 334 „Figure 1“, line 375 „Figure7“, line 380 „Figure8“, line 407 „Fig.9“, line 422“Fig 10“.

In Figure 6, the lines for distinguishing between models and years are too long.

It must always be a space between the numerical value and unit symbol except the plane angle and percent – line 352 „50MWth“.

The reference list needs to be adjusted – line 583 „11. B, H.D.A., et al“, line 585 „12. B, W.W.Z.A., et al.“. Very suspicious authors!

Author Response

(1) The description of the models used in the analysis is practically very general. The relationships and calculation equations underlying the models are completely missing from the article. The assumptions and limitations and possible uncertainties of the models used have not been analyzed in detail. It also does not make it possible to evaluate the results obtained.

Answer:We refined the framework of the model, introduced the description and linkage of each model. We have added some formulas and content about the CGE model, GAINS model, Health model in the method chapter. The scenario assumptions have been described, and the effects of parameter acquisition of exposure response function and the selection of three methods on the results are compared. The exposure response function has a certain change on the linear and nonlinear GEMM method. The analysis of uncertainty is conducive to bring some certainty to the results and increase the reliability of the results.

(2) The technical implementation of the article does not meet the standards of academic writing. Academic writing should be objective. If it is subjective or emotional, it will lose persuasiveness and may be regarded as relying on emotion rather than building a reasonable argument based on evidence. The language or informal writing should therefore be impersonal, and should not include personal pronouns. For most subject areas the writing is expected to be objective. For this the first person (I, we, me, my, etc.) should be avoided. In this article on line 13 is written “ we construct", line 16 „We analyze“, line 146 „we identify“, line 153 „we build“, line 162 „We then estimate“, line 205 „we can use“, line 237 „we use“, line 238 „we compare“, line 250 „we obtain“, line 311 „We set“, line 510 „we also consider“ and „we find“, line 519 „We evaluate“, line 527 „we consider“, line 528 „we also need“ and line 541 „we find“. Eliminating personal pronouns from writing is highly recommend.

Answer:We have checked all parts of the full text that contain the first person (I, we, me, mine, etc.), using passive voice descriptions, and eliminating personal pronouns from writing. A more objective description of the facts has been changed into the passive voice, as shown in the text.

Specific comments:

(3)The same style should be used throughout the article, either PM2.5 (lines 3, 11, 13, 29, 30, 57, 63, 71, 77, 83, 110, 137, 146, 157, 158, 160, 162, 203, 236, 239, 242,243, 245, 247, 252, 264, 269, 304, 327, 328, 331, 334, 440, 538 and 540) or PM2.5(lines 226, 248, 274, 277, 280, 292, 284, 287, 295, 316, 317, 344, 346, 351, 357, 360, 362, 363, 364, 365, 370, 372, 385, 388, 390, 392, 394, 434, 448, 465, 512, 513, 517, 519 and 542).

Answer:We have unified all the PM2.5 formats that appear in the text, making them all standardized as "PM2.5".

(4)Formulas for chemical substances must be formulated correctly using subscripts – lines 41, 42 „CO2“, line 74 „SO2“, line 75 „NO2“, line 82 „NO2“, line 91 „O3“, line 203 „SO2, NOx“, line 204 „NH3“.

Answer:We have also corrected CO2, SO2, NOX, O3, and NH3, and all are unified into the standard format" CO2、SO2、NOX、O3、NH3",

(5)Units need to be formatted correctly using superscripts– lines 63, 65, 71, 247, 278, 280, 285, 317, 331, 335, 336, 337, 338 and 339.

Answer:We have also corrected. A detailed inspection of the superscripts that appear in the text, such as “ug/m3”,

(6)References to sources must be in square brackets and must not be a superscript – lines 155, 156, 157, 175, 235, 240, 259, 265, 274, 276, 285, 302, 437 and 443.

Answer:A unified format correction to the references used in the text, refer to the format of this journal "[1] ".

(7)All Figures and Tables must be referenced in the text. All Figures and Tables should be inserted into the main text close to their first citation and must be numbered following their number of appearance (Figure 1, Figure 2, Table 1, etc.). There is complete confusion in this article.

When referring to Figures, the same style should be used in the article – line 303 „figure 4“, line 334 „Figure 1“, line 375 „Figure7“, line 380 „Figure8“, line 407 „Fig.9“, line 422“Fig 10“.

Answer:All Figures and Tables has be referenced in the text.  The Figs and tables that appear in the text are unified. The Figs that appear in the text are all unified to “Figure X”, and all Tables are unified to “Table X”, and the serial number is modified to make it correct.

(8)In Figure 6, the lines for distinguishing between models and years are too long.

Answer:The abscissa problem in Figure 6 has been modified, so that the positions of the year and the scene have been reversed.

(9)It must always be a space between the numerical value and unit symbol except the plane angle and percent – line 352 „50MWth“.

Answer:We have add a space between the numerical value and unit symbol except the plane angle and percent – line 352 „50 MWth“.

(10)The reference list needs to be adjusted – line 583 „11. B, H.D.A., et al“, line 585 „12. B, W.W.Z.A., et al.“. Very suspicious authors!

Answer:We have check all the reference, and adjust to the right format

Reviewer 3 Report

This paper was improved

Author Response

(The authors gave the same response as above.)

Reviewer 4 Report

The present manuscript reports a study on the economic impacts of GHG emission and air pollution in the Guangdong province, China. Analyses of current trends and projects, obtained with an integrated modelling scheme, are reported:

  • Line 86: one or more references must be provided for this statement. I suggest the following: doi:10.3390/resources8010015
  • Line 93: a reference must be provided for this statement.
  • Lines 287-293: this paragraph is not clear and should be better formulated. What do the authors mean for “because of the difference in statistical calibre or range”?. Why in the following sentence they repeat the aim of the study?
  • Figure 3 caption. The averaging period of the reported values must be specified
  • Figure 4 legend, replace “labor” with “labor force” or similar
  • Lines 372-373: one or more references must be provided for this statement.
  • Figure 7 caption. Replace current text with “Hospital admissions and mortality cases caused by PM2.5 concentration”
  • Line 442: define the meaning of acronym GEMM

Author Response

The present manuscript reports a study on the economic impacts of GHG emission and air pollution in the Guangdong province, China. Analyses of current trends and projects, obtained with an integrated modelling scheme, are reported:

(1)Line 86: one or more references must be provided for this statement. I suggest the following: doi:10.3390/resources8010015

Answer: Line 86: we have add this reference provided for this statement.

(2)Line 93: a reference must be provided for this statement.

Answer:  Line 93: we have add a reference provided for this statement

(3)Lines 287-293: this paragraph is not clear and should be better formulated. What do the authors mean for “because of the difference in statistical calibre or range”?. Why in the following sentence they repeat the aim of the study?

Answer:We have revised this passage and restated it in page 9.

“The PM2.5 concentration data from NASA satellite data and ground monitoring stations show the concentration changes from 2000 to 2017 in Figure 3. At the same time, the Environmental Bulletin of Guangdong Province has reported the environmental quality of Guangdong Province since 2012 shown in Figure 3[2, 29-33], based on comprehensive statistics data from multiple monitoring points. In the past, due to the lack of local monitoring data, PM2.5 in Guangdong Province was approximately 17 μg/m3 around 2000. With rapid economic growth, energy consumption gradually increased, and the concentration of PM2.5 has risen, reaching 35 μg/m3 by 2008. Then, Guangdong Province began vigorously promoting reduction technology set up, and PM2.5 concentrations began decreasing. But according to the Environmental Statistics Bulletin data of Guangdong Province, PM2.5 decreased from 47 μg/m3 in 2012 to about 31 μg/m3 in 2017[34].” 

(4)Figure 3 caption. The averaging period of the reported values must be specified

Answer:In order to more clearly explain the monitoring concentration data of PM2.5 from 2012 to 2017, the monitoring concentration of the five years was averaged to form a linear straight line. In order to avoid ambiguity, the study removed this simulated average curve. Due to the availability of PM2.5 concentration monitoring data and the accuracy of further research, this study selected the monitoring values in 2015 as the benchmark value, comparison value, reference value and health impact assessment value for future model predictions.

(5)Figure 4 legend, replace “labor” with “labor force” or similar

Answer:We have replace all the “labor” with “labor force” 

(6)Lines 372-373: one or more references must be provided for this statement.

Answer:Line 372-373: We have add a reference provided for this statement

(7)Figure 7 caption. Replace current text with “Hospital admissions and mortality cases caused by PM2.5 concentration”

Answer:We have replace Figure 7 text with “Hospital admissions and mortality cases caused by PM2.5 concentration”

(8)Line 442: define the meaning of acronym GEMM
Answer:We have define the GEMM with (Global Exposure Mortality Model)

Reviewer 5 Report

  1. Inconsistent writing of references, pay close attention to unification. For example, list all author names.
  2. Ref [15] is also about CGE model (L 82 [10-14])
  3. Add references corresponding to L 96-98 “some studies”.

  1. There are many errors in the figure numbers, so correct them.

L 262 Figure 1 -> Figure 5

L 361 Figure 1 -> Figure 11

L 378 Figur2 2 -> Figure 12

  1. Since this study target is Guangdong Province, 3.1-3.3 should be in the methodology( Chapter 2).

  1. 3.4 parameter

 All parameter sets used in CGE model, GAINS model, and Health Impact Module should pe presented in the appendix. This is indispensable because it proves that the three models have been coupled.

  1. Results

L 262-264 : The reliability of the calculation and its width should be added in Figure 5.

Figure 7: It is unclear which item the two axes of the bar graph correspond to.

L 303-305 : It doesn't look like "significantly reduced" and is not a scientific description.(Figure 8)

 L 329-330 : It is unclear which item the transportation sector refers to.(Figure 9)

 L 359-361 : Due to the large error bars, the tendency is not clear unless it shows a statistically significant difference. Should be carefully considered and corrected.

  1. Discussion

 L 402-410 : The description of this part is only the analysis result and is not controversial. It needs to be completely rewritten.

The discussion requires comparison with other similar studies, so clarify where new findings were obtained in this paper. You can combine it with 5.3.

  1. Conclusions

 L 430-431 : Show who made such a proposal or suggestion.

Gain or a net benefit of 0.11 % : Since the same expression is found in two places, please avoid duplication and rewrite it..

Author Response

(1) Inconsistent writing of references, pay close attention to unification. For example, list all author names.

Ref [15] is also about CGE model (L 82 [10-14])

Add references corresponding to L 96-98 “some studies”.

Answer:We rearrange the references of the whole article to make the context corresponding and the format unified. Some studies and some missing references are supplemented.

(2) There are many errors in the figure numbers, so correct them.

L 262 Figure 1 -> Figure 5

L 361 Figure 1 -> Figure 11

L 378 Figur2 2 -> Figure 12

Answer:We rearrange the Figure numbers in this paper to unify the context.

(3) Since this study target is Guangdong Province, 3.1-3.3 should be in the methodology ( Chapter 2).

3.4 parameter

All parameter sets used in CGE model, GAINS model, and Health Impact Module should be presented in the appendix. This is indispensable because it proves that the three models have been coupled.

Answer:We moved sections 3.1-3.3 as database to Section 2. We added the formula descriptions of CGE model and gains model. The parameter tables of the models and the input and output data between the models are placed in the table in the Appendix.

Appendix:  Table 1. Exposure-Response Functions

Impact Category

ERFs

C.I. (95%).

Low

C.I. (95%)

High

Chronic mortality

0.40%

0.03%

0.80%

Respiratory hospital admissions

1.17E-05

6.38E-06

1.72E-05

Cerebrovascular hospital admission

8.40E-06

6.47E-07

1.16E-05

Cardiovascular hospital admissions

7.23E-06

3.62E-06

1.09E-05

Chronic bronchitis

4.42E-05

-1.82E-06

9.02E-05

Asthma attacks

1.22E-04

4.33E-05

12.08E-04

Respiratory symptoms days

2.50E-2

2.17E-1

4.05E-1

work loss day

2.07E-02

1.76E-02

2.38E-02

Table.2 Energy consumption for each sector in Guangdong (Mtce)

2020

2025

2030

2035

Agriculture

5.59

6.56

6.09

5.65

Power generation

48.87

48.91

46.50

44.22

Petroleum mining

0.06

0.01

0.00

0.00

Natural gas mining

2.95

3.03

1.63

0.88

Other mining

1.67

1.97

2.04

2.11

Food manufacturing

5.34

5.97

5.72

5.48

Textile

5.04

5.27

4.42

3.71

Wood processing

1.69

1.76

1.69

1.64

Papermaking

9.44

10.28

10.91

11.59

Other manufacturing

6.71

7.12

6.65

6.22

Oil refining

12.09

14.44

16.29

18.38

Coking

11.53

8.83

2.69

0.82

Chemical industry

27.93

33.32

37.78

42.85

Cement

11.92

12.44

12.67

12.90

Other non-metallic manufacturing industry

10.59

11.78

10.13

8.71

Glass manufacturing

4.18

4.50

3.70

3.04

Ceramic manufacturing

3.47

3.97

3.65

3.37

Steel

74.75

89.20

106.91

128.13

Non-ferrous metal smelting

7.43

9.04

9.86

10.76

Metalware

8.95

10.03

10.22

10.41

Mechanical manufacturing

11.09

12.27

12.78

13.30

Electronic equipment manufacturing

16.51

15.57

13.79

12.21

Gas production and supply

1.01

1.12

1.01

0.92

Water production and supply

2.00

2.29

2.52

2.77

Construction

3.88

4.43

4.54

4.65

Road transport

1.81

2.17

2.22

2.28

Railway transportation

9.38

11.58

11.71

11.84

Urban public transport

7.39

10.48

10.66

10.84

Water transport

10.64

12.59

11.45

10.41

Air transport

11.17

13.40

15.64

18.24

Other transportation

0.07

0.08

0.07

0.06

Service industry

18.06

22.90

25.35

28.05

Total

353.20

397.31

411.30

436.42

(4) Results

L 262-264 : The reliability of the calculation and its width should be added in Figure 5.

Answer:Thanks the reviewers for this questions. In Figure 5, we built a scenario from CGE model to predict future energy consumption and pollutant emissions in WOC scenario. According to the CGE model, the fossil energy consumption is imported into GAINS model, and the future concentration of PM2.5 is calculated according to the fossil energy consumption level and the background value of PM2.5 concentration in other parts of China in 2015.

Further, according to the gains model, we designed three different scenarios to predict the pollutant concentration. The pollutant concentration in different scenarios presents an annual average value. The concentration is different under different scenarios, representing an upper and lower range based on the different policy.

(5) Figure 7: It is unclear which item the two axes of the bar graph correspond to.

Answer: We have modified this diagram.

(6) L 303-305 : It doesn't look like "significantly reduced" and is not a scientific description.(Figure 8)

Answer: We deleted the "significantly”.

(7) L 329-330 : It is unclear which item the transportation sector refers to.(Figure 9)

Answer:We have added a sentence to explain that the transportation sector is divided into railway, waterway, highway, aviation and pipeline.

(8) L 359-361 : Due to the large error bars, the tendency is not clear unless it shows a statistically significant difference. Should be carefully considered and corrected.

Answer:Error bar indicates that the factors of exposure response function under different scenarios. The parameters of exposure response have certain upper and lower limits, and there is a certain influence range from concentration to pathogenicity. The main purpose of this study is to analyze the macroeconomic impact, and take the median of this factor as the parameter to evaluate the economic impact. The uncertainty of this deviation is within 60%.

(9) Discussion

L402-410 : The description of this part is only the analysis result and is not controversial. It needs to be completely rewritten.

Answer:We compared the results of such studies, considering the impact of pollutants on economic losses and the literature on the positive impact of pollutant emission reduction on economy. At present, due to the different regional differences, there are great differences in the impact assessment of the cost and benefits of pollutant treatment technology, which is related to the local industrial, technical and economic structure. This paper gives a health impact analysis through a case study and a study of Guangdong as the largest province with a large population. As the province with the largest GDP and the largest population in China, it will provide quantitative data support for the formulation of environmental protection policies and public environmental governance in the future.

We add the new discussion in the text.

“Compared with other scholars [5-7], some have considered the positive externali-ties brought by the treatment of pollutants and come to the conclusion that the impact on the local regional economy is positive. Other scholars [12,14] did not consider the macro-economy of the system and concluded that pollution control brought about large investment costs, or air pollution brought about economic losses of 1-3%. This study described the whole life cycle of energy consumption to support economic growth while produce air pollutant on environment and health. The analysis of the cost and economic benefits of controlling air pollutants will improve the cost effec-tiveness of the government's action of formulating emission reduction policies and persuading public finance to make great efforts to control air pollution, which is a pub-lic product, and investment on pollutant treatment can bring benefits.  ”

 (10) The discussion requires comparison with other similar studies, so clarify where new findings were obtained in this paper. You can combine it with 5.3.

Answer:Thank you for your good suggestions. We have merged with 5.3 and compared the research results with other domestic and foreign literature [5-7,12,14]. Because Guangdong is a coastal province with developed industry and economy, but at the same time, the air quality diffusion is good, and the concentrations of PM2.5 and other pollutants are relatively clean. Our research found that in the short term, the cost of air pollutant treatment will be lower than the benefit, but with the deepening of governance, the gradual health benefit will be greater than the costs. At present, many health costs are underestimated or can’t be considered in environmental governance investment decisions. Through the CGE prediction model, this study reveals that in the areas with serious environmental pollution, the improvement of human health and labor quality are effective by controlling pollutants.

(11) Conclusions

L 430-431 : Show who made such a proposal or suggestion.

“We evaluate the health impact of PM2.5 pollution in Guangdong Province from 2015 to 2035. The results show that the increase in outpatient numbers, inpatient numbers and early death rate would increase medical costs and work loss time.

Gain or a net benefit of 0.11 %: Since the same expression is found in two places, please avoid duplication and rewrite it.

Answer:Thank you for your question. We have revised this sentence. Through research, we come to the conclusion that the cost and benefit of pollution have different returns under different scenarios, and the costs and benefits in various sectors are also different. It is suggested that the government should implement and supervise environmental governance measures. Especially when financial funds are limited, this study can give priority to implement policies and increase the energy or transport sector transformation and support.

We add the sentences in the text.

“The results shows that strengthening air pollution control can reduce PM2.5 pollution, reduce the impact on health, and the improvement of labor health quality, including the reduction of employment time and labor expenditure, promote the overall economic growth. Considering the cost of control technology, the improvement in PM2.5 pollution in 2035 will bring a net benefit of 0.11% of GDP.”

Round 2

Reviewer 2 Report

The review on the manuscript in journal Sustainability entitled „Health and economic impact assessment of Transport and Industry PM2.5 control policy in Guangdong Province“.

The article analyzes the additional damages related to PM2.5 pollution (additional medical expences, mortality, etc.) in Guangdong Province under different scenarios and ways to reduce them.

The article has been significantly supplemented and corrected.

Research methods have been described at satisfactory level.

The conclusions are based on analysis and are adequate.

The article needs a minor technical corrections.

Specific comment

Units need to be formatted correctly using superscripts– lines 64 and 348.

In many places, there are no spaces between words – lines 66, 71, 86, 103, 104, 108, 282, 291, 320, 342, 365 and 516.

Line spacing 193 - 214 is too large.

The "where" following a equation is part of the sentence containing the equation and must begin with a lowercase letter - lines 199, 205, 249, 262 and 301.

When referring to websites, I must provide the following information - Title of Site. Available online: URL (accessed on Day Month Year). Such information is not available for reference [1].

Author Response

Thank you very much for the valuable comments. They are quite helpful to improve our study.

Specific comment

1. Units need to be formatted correctly using superscripts– lines 64 and 348.

Answer:We have corrected the lines 64 and 348, “ ug/m3 ” to be formatted correctly “ug/m3

2. In many places, there are no spaces between words – lines 66, 71, 86, 103, 104, 108, 282, 291, 320, 342, 365 and 516.

Answer:We have added the spaces between words– lines 66, 71, 86, 103, 104, 108, 282, 291, 320, 342, 365 and 516.

3. Line spacing 193 - 214 is too large.

Answer:We have adjust the line spacing between the line 193 - 214.

4. The "where" following a equation is part of the sentence containing the equation and must begin with a lowercase letter - lines 199, 205, 249, 262 and 301.

Answer:We have change the “Where” in lines 199, 205, 249, 262 and 301 to “where”.

5. When referring to websites, I must provide the following information - Title of Site. Available online: URL (accessed on Day Month Year). Such information is not available for reference [1].

Answer:We have change the reference to
“XINHUANET, http://www.xinhuanet.com/english/2020-12/12/, Full Text: Remarks by Chinese President Xi Jinping at Climate Ambition Summit. 2020.”

Reviewer 5 Report

Manuscript is well modified.

Author Response

Thank you very much for the valuable comments. They are quite helpful to improve our study.

This manuscript is a resubmission of an earlier submission. The following is a list of the peer review reports and author responses from that submission.

Round 1

Reviewer 1 Report

The provided manuscript is about different air pollution control strategies scenarios modelled in Guangdong Province. 

Main remarks:

  • The manuscript is not well-organized. Some sentences are not very clear. 
  • Almost all figures are not cited in the text. Reference citation numbers should be placed inside the punctuation.
  • Some abbreviated phrases aren't explained or written in full (WTP, VSL, HCA, COI, etc.).
  • A bit strange use of the term department (Table 1). It is not clear on what basis "departments" are classified. 
  • All figures are not correctly constructed (no axis titles, it is not clear the units, etc.). 
  • Subsections 4.2 is too short, with only one sentence.
  • Could be described in more detail what air control measures are encountered for the modelling, what measures encountered for Blue sky strategy. 

Reviewer 2 Report

The review on the manuscript in journal Sustainability entitled „Health and economic impacts assessment of PM2.5 control policy in Guangdong province“.

The article analyzes the additional damages related to PM2.5 pollution (additional medical expences, mortality, etc.) in Guangdong province under different scenarios and ways to reduce them.

Unfortunately, the research method is presented in a confusing and not complete way. Also, the presentation of research results does not meet the generally accepted standards of academic writing.

Broad comments

The description of the models used in the analysis is practically non-existent or very general. The assumptions and limitations and possible uncertainties of the models used have not been analyzed in detail.

The structure and technical implementation of the article is particularly poor and does not meet the minimum standards of academic writing.

Figures and Tables should be numbered in the order they appear in the text. All Figures and Tables need to be referenced in the text. Figures and Tables should be referenced in the order they appear in the text (i.e. Figure 1 is referenced in the text before Figure 2 and so forth). The article refers first to Figure 3 (p. 8), then to Figure 6 (p. 9), then to Figure 1 (p. 12) and lastly to Figure 2 (p. 13). There is no references to Figures 5, 7, 8, 9, 10, 11 and 12 in the article. The article on page 12 contains a Table without a number. The article refers to Tables 1 (page 3) and 2 (page 14), the rest are not referenced in the article.

The titles of the Figures must be only under Figure, the titles do not need to be repeated in the Figure graphic area (Figure 2, Figure 3, Figure 4, Figure 10).

Figures 2, 3, 4, 6, 7, 8, 9 and 10 do not have vertical and/or horizontel axis titles and units.

The article needs major technical revision.

Specific comments

Academic writing should be objective. If it is subjective or emotional, it will lose persuasiveness and may be regarded as relying on emotion rather than building a reasonable argument based on evidence. The language or informal writing should therefore be impersonal, and should not include personal pronouns. For most subject areas the writing is expected to be objective. For this the first person (I, we, me, my, etc.) should be avoided. In this article on line 12 is written “ we constructed …", etc. Eliminating personal pronouns from writing is highly recommend.

The same style should be used throughout the article, either PM2.5 or PM2.5.

A space must always be placed between the word and the parentheses – line 13 „Equilibrium(CGE)“ etc., etc.

A space must always be placed after the comma – line 19 „15%,the“, etc., etc.

There is no space between the word and the comma - line 23 „( 1.2%) , textile“

Formulas for chemical substances must be formulated correctly using subscripts – line 36 „CO2“, correct „CO2“.

Units need to be formatted correctly – line 407 „30 μ g/m3“, must be „30 μg/m3“.

Line 42 „kW. [1]In“ - A reference to the sentence is placed before the end dot of the sentence. The sentence does not begin with a reference. Etc., etc.

It must always be a space between the numerical value and unit symbol except the plane angle and percent – line 52 „33μg/m3“, etc., etc.

Subheading titles must start with a capital letter – lines 266, 354, 364.

Captions of the Figures must begin with a capital letter – lines 229, 278, 280, 312, 352.

The quality of Figure 2 is very poor.

The reference list needs to be adjusted – line 418 „net, x., Full Text:“ ?; line 438 „B, H.D.A., et al.,“ ?; line 440 „B, W.W.Z.A., et al.“ ?

Reviewer 3 Report

This paper is suitable for publication after minor revision. Please check, if all your figures are mentioned in the text of the paper .